# Large domain movements through the lipid bilayer mediate substrate release and inhibition of glutamate transporters

Xiaoyu Wang[1†], Olga Boudker[1,2†]*

[1]Department of Physiology and Biophysics, Weill Cornell Medicine, New York, United States; [2]Howard Hughes Medical Institute, Chevy Chase, United States

**Abstract** Glutamate transporters are essential players in glutamatergic neurotransmission in the brain, where they maintain extracellular glutamate below cytotoxic levels and allow for rounds of transmission. The structural bases of their function are well established, particularly within a model archaeal homolog, sodium, and aspartate symporter $Glt_{Ph}$. However, the mechanism of gating on the cytoplasmic side of the membrane remains ambiguous. We report Cryo-EM structures of $Glt_{Ph}$ reconstituted into nanodiscs, including those structurally constrained in the cytoplasm-facing state and either apo, bound to sodium ions only, substrate, or blockers. The structures show that both substrate translocation and release involve movements of the bulky transport domain through the lipid bilayer. They further reveal a novel mode of inhibitor binding and show how solutes release is coupled to protein conformational changes. Finally, we describe how domain movements are associated with the displacement of bound lipids and significant membrane deformations, highlighting the potential regulatory role of the bilayer.

*For correspondence:
olb2003@med.cornell.edu

Present address: [†]Physiology and biophysics, Weill Cornell Medicine, New York, United States

## Introduction

Sodium and aspartate symporter $Glt_{Ph}$ is an archaeal homolog of human glutamate transporters, which clear the neurotransmitter glutamate from the synaptic cleft following rounds of neurotransmission (*Danbolt, 2001*). $Glt_{Ph}$ has served as a model system to uncover the structural and mechanistic features of glutamate transporters (*Yernool et al., 2004*; *Boudker et al., 2007*; *Reyes et al., 2009*; *Reyes et al., 2013*; *Akyuz et al., 2013*; *Akyuz et al., 2015*; *Verdon et al., 2014*; *Scopelliti et al., 2018*; *Erkens et al., 2013*; *Hänelt et al., 2015*; *McIlwain et al., 2016*). Recently, structural studies of the family members, including human variants, have enriched the field and have been mostly consistent with earlier findings on $Glt_{Ph}$ (*Canul-Tec et al., 2017*; *Garaeva et al., 2018*; *Yu et al., 2019*). These studies collectively provide what appears to be a nearly complete picture of the structural changes that underlie transport. Briefly, the transporters are homotrimers with each protomer consisting of a centrally located scaffold or trimerization domain and a peripheral transport domain that harbors the L-aspartate (L-asp) and three sodium ($Na^+$) ions binding sites. The crucial conformational transition from the outward-facing state (OFS), in which L-asp binding site is near the extracellular solution, into the inward-facing state (IFS), from which the substrate is released into the cytoplasm, involves a rigid-body 'elevator-like' movement of the transport domain by ca 15 Å across the lipid membrane (*Reyes et al., 2009*; *Akyuz et al., 2013*; *Erkens et al., 2013*; *Ruan et al., 2017*). The structures of the apo transporters in the OFS and IFS showed similar positions of the transport domains that have undergone local structural rearrangements associated with the release of the bound L-asp and $Na^+$ ions (*Verdon et al., 2014*; *Jensen et al., 2013*).

The OFS and IFS conformations show a remarkable internal symmetry (*Yernool et al., 2004*; *Reyes et al., 2009*; *Crisman et al., 2009*). In particular, the transport domains feature two pseudo-symmetric helical hairpin (HP) 1 and 2. HP1 lines the interface between the transport and scaffold

domains in the OFS, reaching from the transporter's cytoplasmic side. HP2 lies on the surface of a large extracellular bowl formed by the transporter and occludes L-asp and three $Na^+$-binding sites (NA1, 2, and 3). The two hairpins meet near the middle of the lipid bilayer, and their non-helical tips provide essential coordinating moieties for the bound L-asp. As the transport domain translocates into the IFS, HP2 replaces HP1 on the domains interface, while HP1 now lines an intracellular vestibule leading to the substrate-binding site (*Figure 1—figure supplement 1*). Structural and biophysical studies have established that HP2 serves as the transporter's extracellular gate (*Boudker et al., 2007*; *Verdon et al., 2014*; *Focke et al., 2011*; *Riederer and Valiyaveetil, 2019*). HP2 closes when the transporter is bound to $Na^+$ ions and L-asp and when it is empty (*Verdon et al., 2014*; *Yernool et al., 2004*; *Jensen et al., 2013*). In contrast, it assumes open conformations when the transporter is bound only to $Na^+$ ions or $Na^+$ ions and competitive blockers DL-*threo-β*-benzyloxyaspartate (TBOA) or (2S,3S)−3-[3-[4-(trifluoromethyl)benzoylamino]benzyloxy]aspartate (TFB-TBOA) (*Boudker et al., 2007*; *Verdon et al., 2014*; *Canul-Tec et al., 2017*).

The gating process in the IFS is less well understood. Based on symmetry considerations, it was first proposed that HP1 might serve as the intracellular gate (*Yernool et al., 2004*) or that the very tip of HP2 might open to release the substrate and ions (*DeChancie et al., 2011*). A large opening of HP2 seemed unlikely because of the steric constraints on the domain interface. However, later structures of a gain-of-function mutant of $Glt_{Ph}$ and human homologous neutral amino acid transporter ASCT2 showed that the transport domain in the IFS could swing away from the scaffold, opening a crevice between the domains (*Akyuz et al., 2013*; *Garaeva et al., 2018*). In this so-called 'unlocked' conformation, there was sufficient space for HP2 to open. More recent studies of ASCT2 and of an archaeal $Glt_{Tk}$, a close homolog of $Glt_{Ph}$, further showed that HP2 could open, suggesting that it serves as a gate in both the OFS and IFS (*Garaeva et al., 2019*; *Arkhipova et al., 2020*). Here, we report a series of Cryo-EM structures of $Glt_{Ph}$ reconstituted into nanodiscs in the IFS and OFS. We show that the transport domain explores a large range of motions in the IFS to which the bilayer adapts through significant bending. These motions are coupled to local changes in HP2 to mediate variable exposure of substrate-binding sites to the solvent and accommodate ligands of diverse sizes. They also affect the area of the hydrophobic interface between the transport and scaffold domains. When the transporter is bound to non-transportable blockers or $Na^+$ ions only, the area is significantly larger than when the transporter is apo or fully loaded with the substrate and ions. The more extensive interface may contribute to the transport domain's inability to return to the OFS, providing a mechanism of inhibition and coupled transport.

## Results

### Large range of motions of the transport domain in the IFS

In the outward-facing $Glt_{Ph}$ and EAAT1 in complex with blockers TBOA and TFB-TBOA or $Na^+$ ions only, HP2 opens to various degrees, enabling access to the substrate-binding site (*Canul-Tec et al., 2017*; *Boudker et al., 2007*; *Verdon et al., 2014*). To picture gating in the IFS, we imaged the $Glt_{Ph}$ reconstituted into MSP1E3 nanodiscs in the presence of various ligands by single-particle Cryo-EM. Because wild type $Glt_{Ph}$ strongly prefers the OFS in detergent and lipid environments (*Akyuz et al., 2015*; *Huang et al., 2020*; *Ruan et al., 2017*; *Georgieva et al., 2013*; *Hänelt et al., 2013*), we used a variant of $Glt_{Ph}$, conformationally constrained in the IFS by crosslinking of cysteine residues placed into the transport and scaffold domains, $Glt_{Ph}$-K55C/A364C ($Glt_{Ph}^{IFS}$) (*Reyes et al., 2009*). Earlier crystal structures of $Glt_{Ph}^{IFS}$ pictured the position of the transport domain that was very similar to those visualized in unconstrained inward-facing $Glt_{Ph}$ mutants (*Akyuz et al., 2015*; *Verdon and Boudker, 2012*).

We determined the structures of $Glt_{Ph}^{IFS}$ free of ligands ($Glt_{Ph}^{IFS}$-Apo-open) or in complex with $Na^+$ ions ($Glt_{Ph}^{IFS}$-Na) and bound to L-asp ($Glt_{Ph}^{IFS}$-Asp), TBOA ($Glt_{Ph}^{IFS}$-TBOA), TFB-TBOA ($Glt_{Ph}^{IFS}$-TFB-TBOA), and the wild type outward-facing $Glt_{Ph}$ in complex with TBOA ($Glt_{Ph}^{OFS}$-TBOA) to 3.52, 3.66, 3.05, 3.39, 3.71, and 3.66 Å resolution, respectively (Materials and methods, *Figure 1—figure supplements 2–4*, and *Table 1*). The Cryo-EM $Glt_{Ph}^{IFS}$-Asp structure was nearly identical to the earlier crystal structure (RMSD of 1.0 Å) (*Reyes et al., 2009*). The transport domain was well packed against the scaffold primarily through interactions of HP2 and the extracellular part of TM8 (TM8a) with the scaffold TMs 2, 4, and 5. The central axis of the roughly cylindrical transport domain formed a ~ 35 °

**Table 1.** Cryo-EM data collection, refinement and validation statistics.

| | $Glt_{Ph}^{OFS}$-TBOA (EMD- 21991) (PDB- 6 × 17) | $Glt_{Ph}^{IFS}$-Asp (EMD- 21989) (PDB- 6 × 15) | $Glt_{Ph}^{IFS}$-TBOA (EMD- 21990) (PDB- 6 × 16) | $Glt_{Ph}^{IFS}$-TFB-TBOA (EMD- 21988) (PDB- 6 × 14) | $Glt_{Ph}^{IFS}$-Na (EMD- 21987) (PDB- 6 × 13) | $Glt_{Ph}^{IFS}$-Apo-open (EMD- 21986) (PDB- 6 × 12) |
|---|---|---|---|---|---|---|
| **Data collection and processing** | | | | | | |
| Magnification | 22500x | 130000x | 22500x | 22500x | 22500x | 22500x |
| Voltage (kV) | 300 | 300 | 300 | 300 | 300 | 300 |
| Electron exposure (e–/Å$^2$) | 68.55 | 69.30 | 69.70 | 68.70 | 68.55 | 68.55 |
| Defocus range (μm) | −1.5 to −2.5 | −1.5 to −2.5 | −1.5 to −2.5 | −1.5 to −2.5 | −1.5 to −2.5 | −1.5 to −2.5 |
| Pixel size (Å) | 1.07325 | 1.0605 | 1.07325 | 1.07325 | 1.07325 | 1.07325 |
| Symmetry imposed | C3 | C3 | C3 | C3 | C1 | C1 |
| Initial particle images (no.) | 426089 | 445791 | 1378438 | 1326573 | 962164 | 962164 |
| Final particle images (no.) | 88961 | 74233 | 47950 | 75555 | 191349 | 148582 |
| Map resolution (Å) FSC threshold | 3.66 0.143 | 3.05 0.143 | 3.39 0.143 | 3.71 0.143 | 3.66 0.143 | 3.52 0.143 |
| Map resolution range (Å) | 2.6–7.0 | 2.3–4.0 | 2.4–4.5 | 2.4–4.5 | 2.4–7.0 | 2.4–7.0 |
| **Refinement** | | | | | | |
| Initial model used (PDB code) | 2NWW | 3KBC | 3KBC | 3KBC | 3KBC | 3KBC |
| Map sharpening $B$ factor (Å$^2$) | −182.8 | −94.1 | −97.6 | −174.8 | −157.9 | −131.2 |
| Model composition Non-hydrogen atoms Protein residues Ligands | 9393 1245 3 | 10026 1257 54 | 9438 1248 6 | 9486 1239 9 | 3136 417 1 | 3059 407 1 |
| $B$ factors (Å$^2$) Protein Ligand | 40.69 35.78 | 40.94 49.70 | 85.87 84.33 | 46.95 49.95 | 47.52 41.36 | 75.40 73.63 |
| R.m.s. deviations Bond lengths (Å) Bond angles (°) | 0.006 0.918 | 0.005 0.811 | 0.006 0.945 | 0.005 0.848 | 0.005 0.910 | 0.007 0.951 |
| Validation MolProbity score Clashscore Poor rotamers (%) | 1.42 4.19 0 | 1.34 3.62 0 | 1.52 3.97 0 | 1.56 6.63 0 | 1.68 4.80 0 | 1.23 3.01 0 |
| Ramachandran plot Favored (%) Allowed (%) Disallowed (%) | 96.61 3.39 0 | 96.88 3.12 0 | 95.17 4.83 0.31 | 96.84 3.16 0 | 93.49 6.51 0.31 | 97.27 2.73 0 |

angle with the membrane normal (*Figure 1a,b*). HP2 was closed over the substrate-binding site and packing between the transport and scaffold domains left no space for it to open. A similar inter-domain orientation and packing were also observed in a previously solved crystal structure of the occluded apo $Glt_{Ph}^{IFS}$ ($Glt_{Ph}^{IFS}$-Apo-closed, PDB code 4P19, *Figure 1a*; *Verdon et al., 2014*). In the new Cryo-EM structures of $Glt_{Ph}^{IFS}$-Na, $Glt_{Ph}^{IFS}$-Apo-open, $Glt_{Ph}^{IFS}$-TBOA, and $Glt_{Ph}^{IFS}$-TFB-TBOA, approximately the same regions of HP2 and TM8a remained juxtaposed against the scaffold. However, the bulk of the transport domain swung out away from HP2 and the scaffold to different extents (*Figure 1a*) with the largest angle between the transport domain and the membrane normal of ~ 47° in $Glt_{Ph}^{IFS}$-TFB-TBOA (*Figure 1b*). Together, the crystal and Cryo-EM structures define gating mechanisms in $Glt_{Ph}$ on the extracellular and cytoplasmic sides (*Figure 1c, Video 1*). In the OFS, the bulk of the transport domain remains mostly static relative to the scaffold, and the labile HP2 serves as the extracellular gate. In the IFS, HP2 can maintain interactions with the scaffold, while the bulk of the transport domain swings away to allow access to the binding site. Notably, in a crystal structure of a

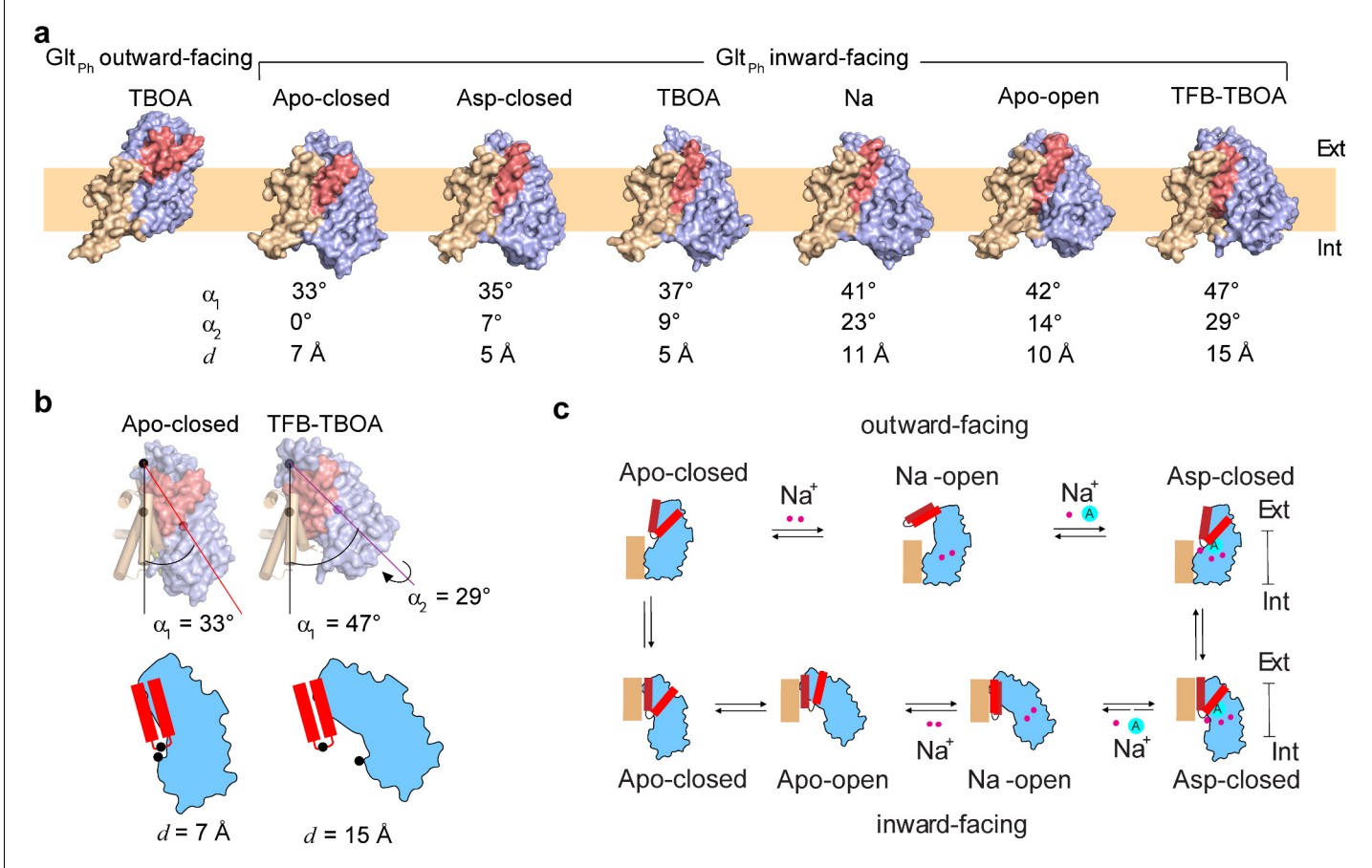

**Figure 1.** Gating mechanism in the IFS. (**a**) Structures of $Glt_{Ph}$ protomers are shown in surface representation viewed in the membrane plane. The scaffold domain is colored wheat, the transport domain blue and HP2 red. The PDB accession code for $Glt_{Ph}^{IFS}$ - Apo-closed is 4P19 (*Verdon et al., 2014*). An approximate position of the bilayer is shown as a pale orange rectangle. (**b**) Angles between the membrane normal drawn through the center of the scaffold domain and the central axis of the transport domains ($\alpha 1$) are shown for $Glt_{Ph}^{IFS}$ -Apo-closed and $Glt_{Ph}^{IFS}$ -TFB-TBOA. Also shown is the rotation angle, $\alpha 2$ of the transport domain in Glt PhIFS -TFB-TBOA relative to $Glt_{Ph}$ IFS - Apo-closed. Distances between the $c_\alpha$ atoms (black circles) of residues R276 and P356 (**d**) are shown for the same structures under the schematic depiction of the transport domains. Corresponding angles and distances are listed under all structures in panel (a). (**c**) A schematic representation of the gating mechanism on the extracellular (top) and intracellular (bottom) sides of the membrane.

The online version of this article includes the following figure supplement(s) for figure 1:

**Figure supplement 1.** Schematic representation of the elevator mechanism of transport by $Glt_{Ph}$.

**Figure supplement 2.** Cryo-EM data processing.

**Figure supplement 3.** Data processing flowchart for $Glt_{Ph}^{IFS}$-TFB-TBOA (**a**) and $Glt_{Ph}^{IFS}$-Na, and $Glt_{Ph}^{IFS}$-Apo-open (**b**).

**Figure supplement 4.** Cryo-EM imaging and data processing validation.

gain-of-function aspartate-bound mutant $Glt_{Ph}^{IFS}$-R276S/M395R, the transport domain is positioned at ∼ 45 ° angle (*Akyuz et al., 2015*), similar to the $Glt_{Ph}^{IFS}$-TFB-TBOA Cryo-EM structure. However, in $Glt_{Ph}^{IFS}$-R276S/M395R, HP2 remains closed over the binding site and a large lipid-filled gap forms between the transport and scaffold domains. It is currently unclear whether the transport domain first swings away from the scaffold providing space for the consequent HP2 opening or whether HP2 remains in place while the bulk of the domain swings out in a 'wag-the-dog' manner.

## Two transporter blockers bind differently to $Glt_{Ph}^{IFS}$

TBOA and TFB-TBOA blockers share the amino acid backbone with L-asp but are decorated on β-carbon with one and two benzyl rings, respectively, that cannot fit within the confines of the

substrate-binding site. They block transport by binding to the outward-facing $Glt_{Ph}$, $Glt_{Tk}$, or EAATs and arresting HP2 in an open conformation (*Boudker et al., 2007*; *Canul-Tec et al., 2017*; *Arkhipova et al., 2020*). Our Cryo-EM structure of the outward-facing $Glt_{Ph}^{OFS}$-TBOA in nanodisc confirmed that the transporter took the same conformation in the absence of crystal contacts in a lipid bilayer (RMSD = 1.0 Å, PDB accession code 2NWW) (*Boudker et al., 2007*; *Figure 2—figure supplement 1a*). TBOA and the related L-β-threo-benzyl-aspartate (TBA) bind to the IFS of $Glt_{Ph}$ (*Reyes et al., 2013*; *Oh and Boudker, 2018*). We used isothermal titration calorimetry to show that TFB-TBOA and TBOA bind to $Glt_{Ph}^{IFS}$ in 200 mM NaCl with 1:1 stoichiometry and the dissociation constants ($K_D$s) of 3.8 and 6.5 µM, respectively (*Figure 2—figure supplement 2a, b*). We then determined the structures of the $Glt_{Ph}^{IFS}$ complexes with the blockers TFB-TBOA and TBOA under saturating conditions in the presence of 10 mM inhibitors.

In the $Glt_{Ph}^{IFS}$-TFB-TBOA structure, TFB-TBOA density was well resolved, and we modeled the inhibitor in its binding site (*Figure 2a*). We also modeled L-asp into the excess density in the binding site of $Glt_{Ph}^{IFS}$-Asp (*Figure 2—figure supplement 1b*). The bound L-asp and TFB-TBOA share some critical interactions (*Figure 2c*). Thus, R397 coordinates the side chain carboxylates of aspartate moieties, and D394 coordinates the amino groups. However, TFB-TBOA assumes a different rotamer, leading to a displacement of the backbone carboxylate and the loss of coordination by the highly conserved N401. The aromatic rings of TFB-TBOA protrude from the ligand-binding site and lodge in between the transport and scaffold domains (*Figure 2a*, *Figure 2—figure supplement 1d*). Most strikingly, HP2 takes a wide-open conformation that is essentially the same as in the outward-facing $Glt_{Ph}$-TBOA complex (*Figure 2—figure supplement 1c*). Interestingly, HP2 was in the same conformation also in an R397C $Glt_{Ph}$ mutant bound to glutamine or benzyl-cysteine. In these structures, the ligands made virtually no interactions with the hairpin but introduced steric clashes disallowing closure (*Scopelliti et al., 2018*). Therefore, it appears that the hairpin intrinsically favors this open conformation.

Surprisingly, HP2 does not open in the same way in $Glt_{Ph}^{IFS}$-TBOA. Instead, the hairpin remains mostly closed, but its N-terminal arm separates from the C-terminal arm. The bound TBOA assumes a similar rotamer as TFB-TBOA, though N401 still coordinates the backbone carboxylate (*Figure 2c*). The C-terminal arm coordinates the sidechain carboxylate of the aspartate moiety as in $Glt_{Ph}^{IFS}$-Asp. The TBOA benzyl group inserts in between the two arms packing against M311 and M362 sidechains (*Figure 2b,c*). The N-terminal arm movement disrupts the Na2 binding site, consistent with previous observations that binding of TBOA and TBA to the IFS of the transporter required only two $Na^+$ ions (*Reyes et al., 2013*; *Oh and Boudker, 2018*). The movement creates a small opening into the cytoplasmic milieu between the tips of HP1 and HP2. It is not clear whether this conformation reflects a functional state. Perhaps, it recapitulates a transient transporter state, in which a $Na^+$ ion has already left the Na2 site while the substrate and two other $Na^+$ ions are still bound. Water might use the cytoplasmic opening to reach and eventually displace the remaining solutes.

These structures collectively show that in $Glt_{Ph}^{IFS}$, bulky competitive blockers can be accommodated either by opening HP2 or by parting its N- and C-terminal arms (*Figure 2d*). Since the OFS and IFS share the same binding pocket for the substrate and competitive inhibitors, it is likely that the new inhibitor binding mode with parted HP2 arms can be sampled in the OFS as well. This mode of blocker binding might provide new pharmacological avenues for the inhibition of human glutamate transporters.

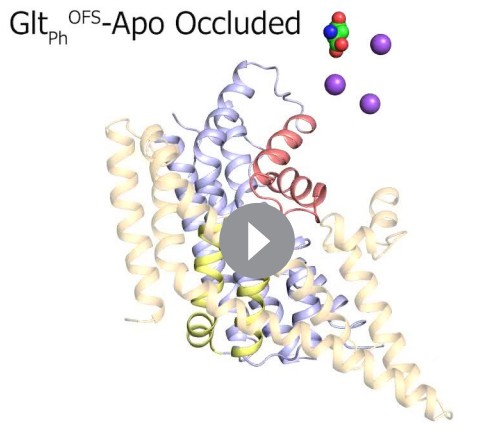

$Glt_{Ph}^{OFS}$-Apo Occluded

**Video 1.** Transport cycle of glutamate transporter $Glt_{Ph}$.
https://elifesciences.org/articles/58417#video1

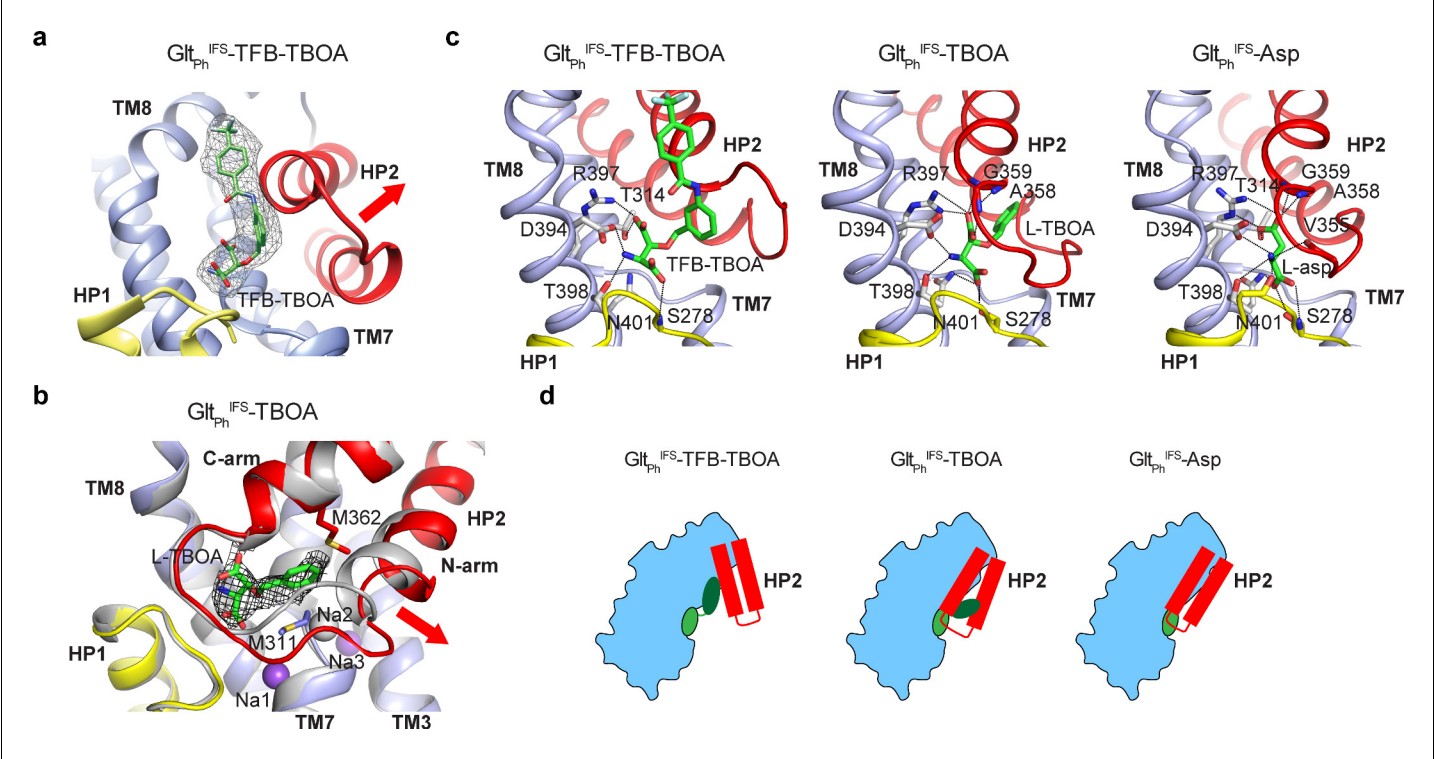

**Figure 2.** Two mechanisms of blocker binding. (**a**) Close-up view of the substrate-binding pocket of $Glt_{Ph}^{IFS}$ with bound TFB-TBOA shown in stick representation and colored by atom type. The corresponding density is shown as a black mesh object. The red arrow emphasizes the HP2 opening. (**b**) Superimposed $Glt_{Ph}^{IFS}$ transport domains in complex with L-asp (gray) and TBOA (colored). The red arrow emphasizes the movement of the N-terminal arm of HP2. TBOA and $Na^+$ ions are shown as sticks and spheres, respectively. The black mesh object is the density contoured at 3 σ. (**c**) Bound TFB-TBOA and TBOA assume similar rotamers, distinct from L-asp, and are coordinated differently. The ligands are shown in stick representations; dotted lines correspond to potential hydrogen bonds. (**d**) Two mechanisms of blockers binding to $Glt_{Ph}^{IFS}$ through either opening of HP2 or parting of the two arms to accommodate the bulky moieties.

The online version of this article includes the following figure supplement(s) for figure 2:

**Figure supplement 1.** $Glt_{Ph}$ Cryo-EM structures in the presence of L-Asp or inhibitors.

**Figure supplement 2.** Crosslinked $Glt_{Ph}$-K55C/A364C binds TFB-TBOA (a) and TBOA (b).

## M311 and R397 couple HP2 gating to ion and substrate binding

To further explore the gating mechanism, we aimed to resolve a structure of $Na^+$ only-bound $Glt_{Ph}^{IFS}$ and imaged nanodisc-reconstituted $Glt_{Ph}^{IFS}$ frozen in the presence of 200 mM NaCl (*Figure 1—figure supplement 3b* and *4*, and *Table 1*). We isolated two distinct structural classes of $Glt_{Ph}^{IFS}$ protomers after symmetry expansion and classification without alignment. The structural heterogeneity was not surprising in retrospect because $Na^+$ concentration in the sample was close to the dissociation constant measured for $Glt_{Ph}^{IFS}$ (*Reyes et al., 2013*). Thus, we observed both $Na^+$-bound ($Glt_{Ph}^{IFS}$-Na) and apo ($Glt_{Ph}^{IFS}$-Apo-open) states. We assigned these states based on the conformations of the conserved non-helical NMD motif (residues 310–312) in TM7, which coordinates $Na^+$ ions in the Na1 and Na3 sites, and TM3, part of the Na3 site (*Figure 3—figure supplement 1a*; *Boudker et al., 2007*; *Guskov et al., 2016*). In particular, the M311 sidechain protrudes toward the L-asp and Na2 sites in $Glt_{Ph}^{IFS}$-Na and $Glt_{Ph}^{IFS}$-Asp structures. In contrast, it flips out toward TM3 in our $Glt_{Ph}^{IFS}$-Apo-open structure and the previous $Glt_{Ph}^{IFS}$-Apo-closed crystal structure (*Verdon et al., 2014*). We did not observe density for $Na^+$ ions in the Na1 and Na3 sites of $Glt_{Ph}^{IFS}$-Na. However, all ion-coordinating residues are positioned similarly to $Glt_{Ph}^{IFS}$-Asp (*Figure 3—figure supplement 1b*). Notably, Na1 is coordinated in $Glt_{Ph}^{IFS}$-Asp, in part, by an occluded water molecule (*Figure 3—figure supplement 1b*). In $Glt_{Ph}^{IFS}$-Na, the water is no longer occluded and is part of an aqueous cavity (*Figure 3a*). We

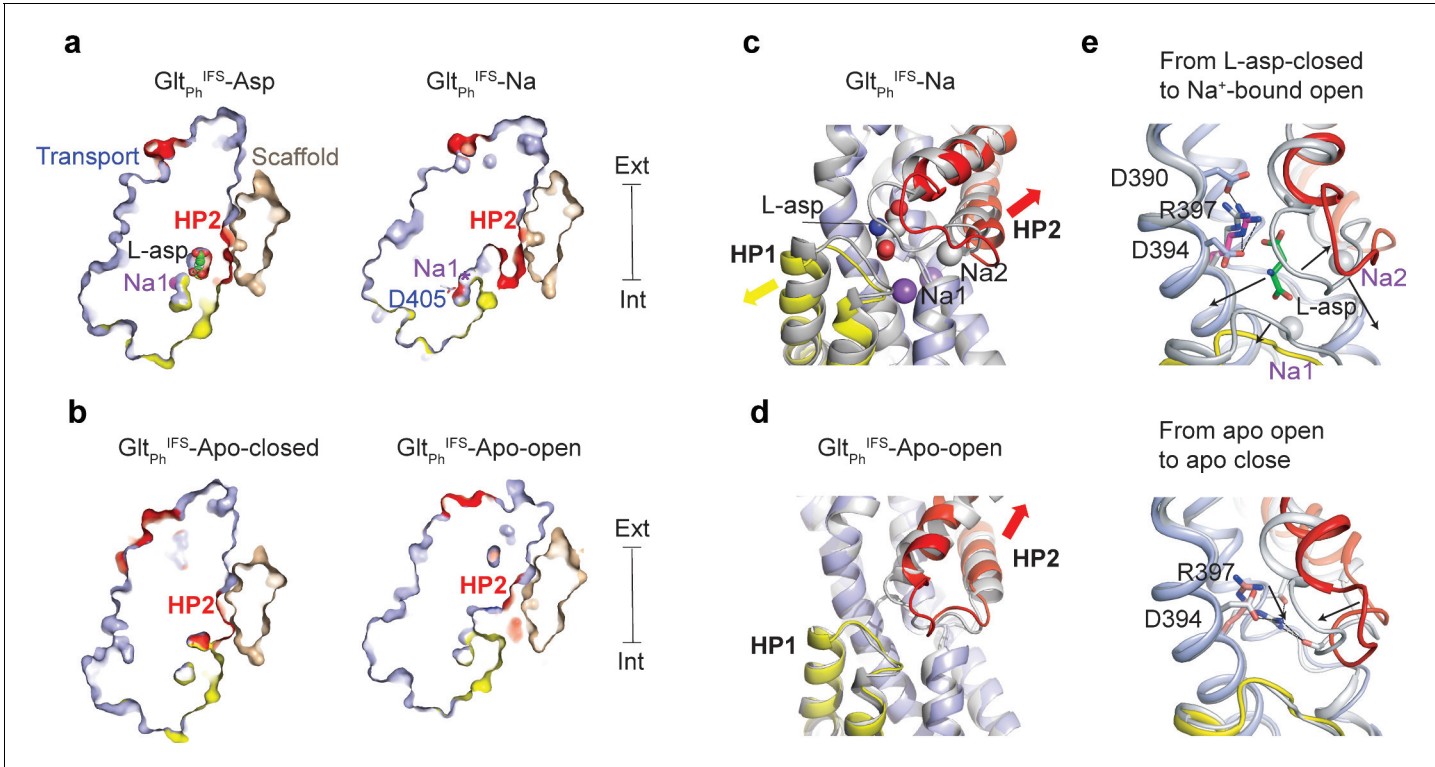

**Figure 3.** Solute-coupled gating. (**a** and **b**) Thin cross-sections of the protomers taken approximately through aspartate-binding sites normal to the membrane plane. The binding site is occluded in Na$^+$/L-asp-bound, and closed apo (PDB 4P19) (*Verdon et al., 2014*) states and is exposed to the solvent in Na$^+$-only, and apo open states. The bound L-asp and Na$^+$ ions in Glt$_{Ph}^{IFS}$-Asp are shown as spheres. In Glt$_{Ph}^{IFS}$-Na, the side chain of D405 is shown as sticks, and a star indicates the Na1 site. (**c**) Superimposed transport domains of Glt$_{Ph}^{IFS}$-Na (colored) and Glt$_{Ph}^{IFS}$-Asp (gray). L-asp and Na$^+$ ions are shown as spheres. Yellow and red arrows indicate movements of HP1 and HP2, respectively. (**d**) Superimposed transport domains of Glt$_{Ph}^{IFS}$-Apo-open (colored) and Glt$_{Ph}^{IFS}$-Apo-closed (gray). (**e**) Gating steps in the inward-facing state. Top: Local structural changes from Glt$_{Ph}^{IFS}$-Asp (gray) to an open Glt$_{Ph}^{IFS}$-Na state (colored). Black arrows indicate the dissociation of L-asp and Na2 and the open states of HP1 and HP2 in Glt$_{Ph}^{IFS}$-Na. Bottom: Binding site occlusion from Glt$_{Ph}^{IFS}$-Apo-open (colored) to Glt$_{Ph}^{IFS}$-Apo-closed (gray). Black arrows mark movements of R397 into the binding site and the closure of HP2.

The online version of this article includes the following figure supplement(s) for figure 3:

**Figure supplement 1.** Na$^+$-binding sites in the Glt$_{Ph}^{IFS}$ in Apo and Na$^+$-bound states.
**Figure supplement 2.** Molecular mechanism of HP2 opening and closing.

conclude that ions likely occupy Na1 and Na3 sites, but the Na1 site might be in rapid equilibrium with the solution.

The Cryo-EM Glt$_{Ph}^{IFS}$-Apo-open structure differs significantly from the occluded Glt$_{Ph}^{IFS}$-Apo-closed crystal structure in that the substrate-binding site is open and hydrated. The opening resembles that in Glt$_{Ph}^{IFS}$-Na compared to the occluded Glt$_{Ph}^{IFS}$-Asp (*Figure 3a,b*) and shares the overall mechanism: HP2 remains in contact with the scaffold while the rest of the transport domain swings out (*Figure 1c*). From the transport domain viewpoint, the conformational changes lead to a similar HP2 opening (*Figure 3c,d*, *Figure 3—figure supplement 2a*). Interestingly, in Glt$_{Ph}^{IFS}$-Na, there is also a small shift of HP1 away from the substrate-binding site, possibly increasing water access to Na1. A similar small movement of the otherwise rigid HP1 was observed in the crystals of apo Glt$_{Ph}^{IFS}$ grown in an alkali-free buffer (*Verdon et al., 2014*).

Two residues in the transport domain - M311 and R397 - move significantly during gating and might couple solute binding and release to large-scale conformational changes. Here we consider a sequence of structural events, which might underlie ion and substrate release in the IFS (*Figure 1c*), starting with Glt$_{Ph}^{IFS}$-Asp and going to Glt$_{Ph}^{IFS}$-Na, Glt$_{Ph}^{IFS}$-Apo-open, and Glt$_{Ph}^{IFS}$-Apo-closed (*Video 2*). In Glt$_{Ph}^{IFS}$-Asp, the R397 side chain extends upward, toward the extracellular side of the membrane, allowing D390 to coordinate its guanidinium group. Thus positioned, R397 makes space for L-asp

and coordinates its sidechain carboxylate, while D394 coordinates its amino group (*Figure 3e*). M311 protrudes into the binding site and coordinates Na2 (*Figure 3—figure supplement 2*). Extensive interaction of HP2 with the bound L-asp and Na2 favor the closed conformation (*Figure 2c*). HP2 opening accompanies L-asp and Na2 release (Glt$_{Ph}^{IFS}$-Na). R397 is now clamped between D390 and D394, while M311 remains in place (*Figure 3e*, *Figure 3—figure supplement 2b*). The consequent release of Na1 and Na3 leads to a restructuring of the NMD motif and outward rotation of M311, which now packs against the open HP2 of Glt$_{Ph}^{IFS}$-Apo-open (*Figure 3—figure supplement 2b*). The guanidinium group of R397 remains between D390 and D394. To achieve the closed apo state, M311 swings further out into the lipid bilayer, allowing HP2 to close. R397 descends deep into the binding pocket, coordinated now only by D394, and is poised to make direct or through-water interactions with carbonyl oxygens of the closed tip of HP2. Steric hindrance of M311 and more positive local electrostatics may prevent R397 from entering the aspartate-binding site and closing HP2 in Na$^+$-only-bound Glt$_{Ph}^{IFS}$. Physiologically, such Na$^+$-bound occluded states should be avoided to prevent Na$^+$ leaks.

Interestingly, in our Cryo-EM analysis, we did not find any Glt$_{Ph}^{IFS}$-Apo-closed structures previously visualized by crystallography. It might be that the open conformation of the apo Glt$_{Ph}^{IFS}$ is the preferred state of the transporter and that the Glt$_{Ph}^{IFS}$-Apo-closed state is assumed only transiently, before the outward transition of the transport domain. Packing crystal contacts, which include extensive interactions between the cytoplasmic sides of the transport domains (*Verdon et al., 2014*), might have stabilized the closed conformation.

## Ligand-dependent domain interface

HP2 and TM8a comprise most of the transport domain surface interacting with the scaffold in Glt$_{Ph}$ inward-facing states. Strikingly, in each of our IFS structures, HP2 takes a different conformation (*Figure 4—figure supplement 1a*). These are similar in structures with occupied Na1 and Na3 sites, that is in complexes with Na$^+$ ions only and with L-asp, TBOA, or TFB-TBOA. The differences are mostly around the tip of HP2 near the L-asp and Na2 sites (*Figure 4—figure supplement 1a and b*). In contrast, the helices restructure significantly in the apo conformations, particularly in Glt$_{Ph}^{IFS}$-Apo-open (*Figure 4—figure supplement 1a and c*). When we superimposed all IFS structures, aligning them on the scaffold domain, we observed that the HP2/TM8a motifs present the same bulky hydrophobic residues flanking the flexible tips for interactions with the scaffold: L347, I361, and L378 form virtually the same spatial arrangement. Only in Glt$_{Ph}^{IFS}$-Apo-open, I350 replaces L347 because the HP2/TM8a motif, particularly the HP2 N-terminal arm, moves outward (*Figure 4—figure supplement 1d and e*).

Thus, the positions of the HP2 tip on the domain interface are mostly conserved. The structural differences in the hairpins then lead to their different orientations relative to the scaffold and different positions of the transport domains, which lean away and rotate to different extents (*Figure 1a, b*). The rotation is small for Glt$_{Ph}^{IFS}$-Asp, relative to Glt$_{Ph}^{IFS}$-Apo-closed (7°), but is significant for Glt$_{Ph}^{IFS}$-Na (23°), and Glt$_{Ph}^{IFS}$-TFB-TBOA (29°) (*Figure 1a,b*). A consequence of these differences is that the bulky residues in the HP2 N-terminal arm, L339, L343, L347, and I350 make more extensive interactions with the scaffold TMs 4a and 4 c in Glt$_{Ph}^{IFS}$-Na and Glt$_{Ph}^{IFS}$-TFB-TBOA compared to other structures. Furthermore, interaction areas between HP2/TM8 and the scaffold domain differ, with Glt$_{Ph}^{IFS}$-Apo-closed and Glt$_{Ph}^{IFS}$-Asp structures having the smallest areas of 1086 and 1076 Å$^2$, respectively, and Glt$_{Ph}^{IFS}$-Na showing the largest increase of ~ 400 Å$^2$ (*Figure 4*).

The interdomain interface disruption is a prerequisite for the transport domain translocation from the inward- to the outward-facing position. Therefore, altered geometry of the interface and larger interaction area may explain why translocation is inhibited by blockers TBOA and TFB-TBOA, or in the transport domain bound to Na$^+$ ions only. While it is not possible to translate interaction areas into energies, it is notable that translocation-competent closed apo and L-asp-bound states show the smallest areas.

Glt$_{ph}^{IFS}$-Asp

**Video 2.** M311 and R397 couple HP2 gating to ion and substrate binding in the inward-facing state.
https://elifesciences.org/articles/58417#video2

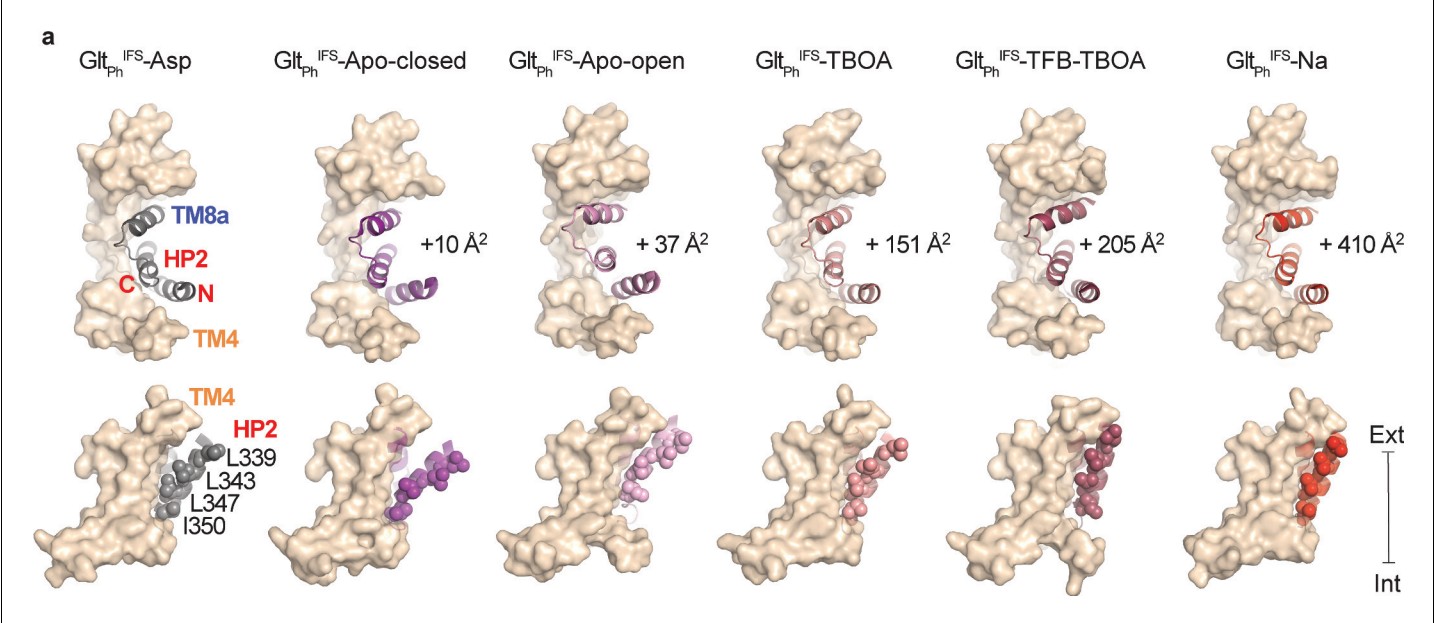

**Figure 4.** Translocation-deficient states show more extensive inter-domain interfaces. (**a**) Surface representations of the scaffold domain in light brown, and cartoon representation of HP2/TM8a motif with Glt$_{Ph}^{IFS}$-Asp colored gray, Glt$_{Ph}^{IFS}$-Apo-closed purple, Glt$_{Ph}^{IFS}$-Apo-open pink, Glt$_{Ph}^{IFS}$-TBOA salmon, Glt$_{Ph}^{IFS}$-TFB-TBOA berry, Glt$_{Ph}^{IFS}$-Na red. Sidechains of L339, L343, L347, and L350 are shown as spheres. Top: viewed from the extracellular space. The increases in the interdomain interaction surface area relative to Glt$_{Ph}^{IFS}$-Asp are shown next to the structures. Bottom: viewed in the membrane plane. Surface areas were determined as described (*Krissinel and Henrick, 2007*). $$BOX_TXT_END$$.

The online version of this article includes the following figure supplement(s) for figure 4:

**Figure supplement 1.** Structural plasticity of HP2 and the inter-domain interface.

Consistently, the crystal structure of the gain-of-function mutant R276S/M395R in the IFS (*Akyuz et al., 2015*) shows a domain interface area of 543 Å², about half of the Glt$_{Ph}^{IFS}$-Asp, and a translocation rate several-fold faster than the wild type transporter.

## Transport domain movements coupled to lipid bilayer

The Cryo-EM structures of the outward- and inward-facing states of Glt$_{Ph}$ are overall similar to the crystal structures. However, they differ in the N-terminus, which is unstructured in crystals but forms a short amphipathic helix positioned on the surface of the nanodiscs in the Cryo-EM OFS and IFS structures (*Figure 1—figure supplement 1*). A similar helix was also observed in crystallized EAAT1 (*Canul-Tec et al., 2017*). We find highly ordered lipid molecules between the N-terminal helix and the rest of the scaffold at positions conserved in all structures (Lipid$_{In}$, *Figure 5a* and *Figure 5—figure supplement 1*). It seems likely that the helix anchors the scaffold domain in the lipid membrane and forms lipid-mediated interactions with the neighboring subunit.

We also find lipid moieties, structured to various degrees, in the crevices between the scaffold and transport domains (*Figure 5a*). Of these, the most notable one is inserted between the N-terminal arm of HP2 and the scaffold TM4a (Lipid$_{Out}$ *Figure 5a,b*). Interestingly, we observe lipids at almost the same location in the Cryo-EM structures of the outward- (*Huang et al., 2020*) and inward-facing L-asp-bound transporters (*Figure 5b*). The lipid packs similarly against TM4a in the OFS and IFS but interacts differently with HP2: near the tip and the extracellular base, respectively. It is not yet clear whether during the outward-to-inward transition, as HP2 slides past TM4a, the lipid is temporarily displaced or disordered. Interestingly, HP2 opening in the OFS, as seen in Glt$_{Ph}^{OFS}$-TBOA, and the IFS requires displacement of Lipid$_{Out}$. Thus, the lipid molecules at this site could modulate gating and the translocation dynamics, affecting both substrate affinity and transport rate. In Glt$_{Ph}^{IFS}$-TFB-TBOA and Glt$_{Ph}^{IFS}$-Apo-open structures, the transport domain leans away from the scaffold far enough to open a window between the two domains that connects the interior of the bilayer

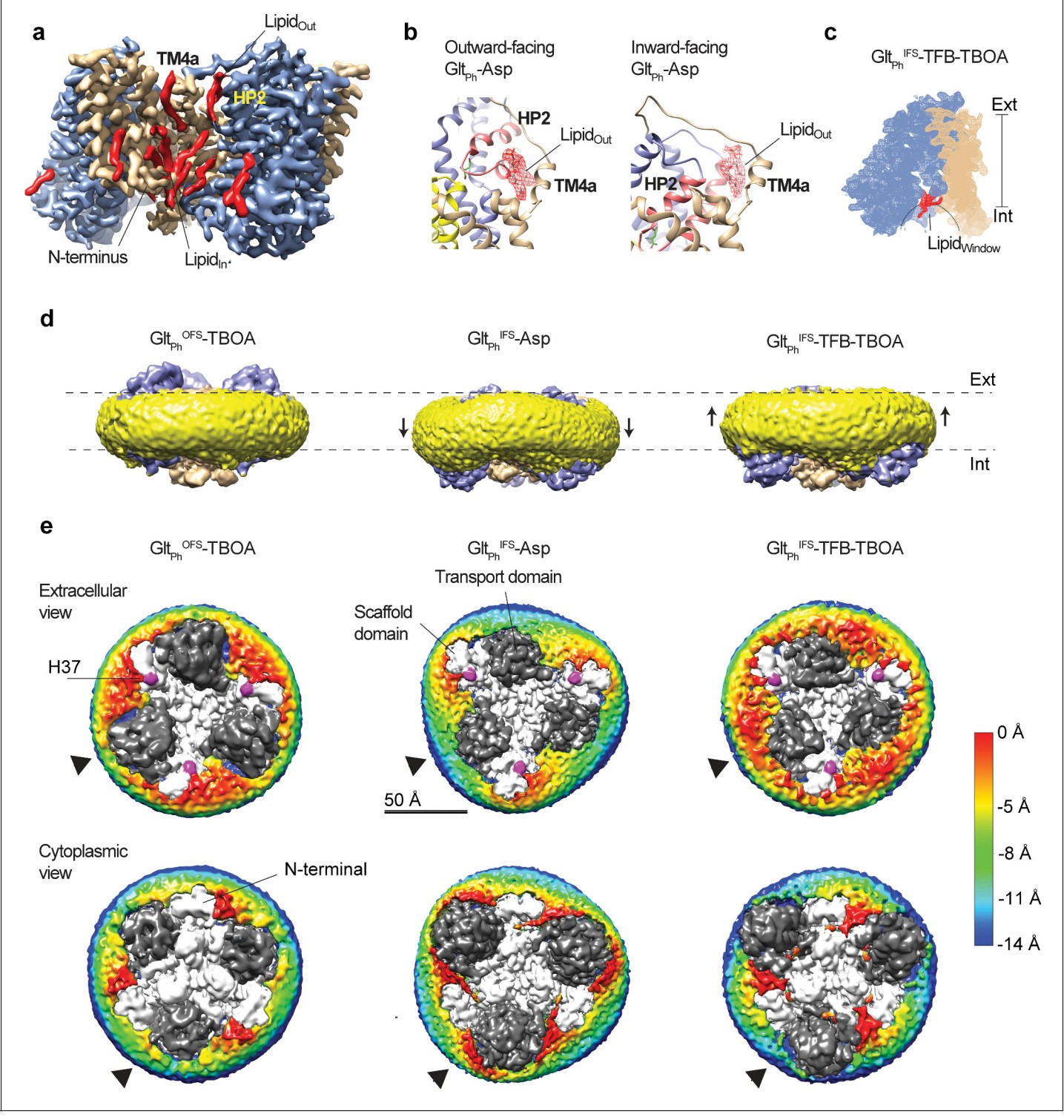

**Figure 5.** Coupling of the lipid bilayer to protein motions. (**a**) Lipid densities (red) observed in protein cervices of $Glt_{Ph}^{IFS}$-Asp. Lipid molecules tucked between the N-terminus and the rest of the scaffold ($Lipid_{in}$) are present in all OFS and IFS structures. (**b**) lipid densities (red mesh objects, $Lipid_{out}$) observed on the extracellular side of a crevice between the scaffold TM4a and HP2 in both outward- (PDB code 6UWF) (***Huang et al., 2020***) and inward-facing $Glt_{Ph}$ bound to L-asp. (**c**) Density map of a $Glt_{Ph}^{IFS}$-TFB-TBOA protomer, with the lipid density in the window between the transport domain and scaffold colored red ($Lipid_{window}$). (**d**) Density maps of $Glt_{Ph}^{OFS}$-TBOA, $Glt_{Ph}^{IFS}$-Asp, and $Glt_{Ph}^{IFS}$-TFB-TBOA in nanodiscs viewed in the membrane plane. Densities corresponding to the transport and scaffold domains are colored blue and wheat, respectively. Density corresponding to the nanodisc is colored yellow. Black arrows mark deviations of the nanodiscs from the planar structures. (**e**) Extracellular (top) and cytoplasmic (bottom) views of the

*Figure 5 continued on next page*

*Figure 5 continued*

density maps. Sections of the maps corresponding to the lipidic nanodisc are colored by their displacement along the membrane normal (scalebar is to the right). The zero-level is set at the surface of nanodisc density around residue H37 (magenta) for the extracellular views. Negative values represent inward bending. The zero-level is set at the surface of nanodisc density around the N-terminus of the protein for the cytoplasmic views. Here, negative values represent outward bending. Densities corresponding to the transport and scaffold domains are shown as dark and light gray, respectively. Black arrows point to the regions with the largest deformations of the nanodiscs observed around the transport domains.

The online version of this article includes the following figure supplement(s) for figure 5:

**Figure supplement 1.** Structured lipids.

to the solvent-filled crevice on the cytoplasmic side of the transporter (*Figure 5c*). We observe excess densities in the opening, suggesting that lipids enter the space at a position structurally symmetric to Lipid$_{Out}$ (Lipid$_{Window}$, *Figure 5c*).

Perhaps most strikingly, we observe nanodisc distortions correlated to positions of transport domains (*Figure 5d,e*, *Video 3*). The nanodisc is nearly flat in the Glt$_{Ph}^{OFS}$-TBOA structure, where the hydrophobic regions of the transport domain and the scaffold are aligned. In Glt$_{Ph}^{IFS}$-Asp, the transport domain forms the sharpest angle to the membrane normal (*Figure 1a*), and its hydrophobic region descends the furthest toward the cytoplasm. The resulting hydrophobic mismatch between the scaffold and transport domains leads to membrane bending to accommodate both, as suggested by recent computational studies (*Zhou et al., 2019*) and studies of Glt$_{Tk}$ (*Arkhipova et al., 2020*). At the protein periphery, the membrane deformation at the transport domain reaches ~ 8 Å shift toward the cytoplasm, observable from both sides of the nanodisc (*Figure 5d,e*, *Video 3*). In contrast, when the inward-facing transport domains swing out, their hydrophobic regions are closer to the extracellular side, and the membrane is less bent. In an extreme case of Glt$_{Ph}^{IFS}$-TFB-TBOA structure, the membrane bends outward, particularly when viewed from the cytoplasmic side (*Figure 5d,e*, *Video 3*). It is unclear whether the nanodisc restricts how far the transport domains swing in the Glt$_{Ph}^{IFS}$-TFB-TBOA structure. Indeed, we observe interactions between the domains and the MSP1E3 lipoprotein, suggesting the size of the nanodiscs might be limiting. Notably, structures of glutamate transporter homologs determined in detergent solutions featured similar positions of the domains (*Garaeva et al., 2019*; *Akyuz et al., 2015*).

## Discussion

The series of structures that we have determined by Cryo-EM suggest that both substrate translocation and substrate gating in the IFS require movements of the transport domain through membrane bilayer. The C-terminal arm of HP2 and TM8a pack against the scaffold near the engineered K55C/A364C crosslink in all IFS structures, while the rest of the transport domain moves to various degrees. It is possible that the crosslink constraints the movements, but we do not think so. First, Na$^+$-bound unconstrained inward-facing Glt$_{Tk}$ (*Arkhipova et al., 2020*) is structurally similar to Glt$_{Ph}^{IFS}$-Na (overall RMSD = 0.7), with little difference in the crosslink region (*Figure 3—figure supplement 2c*). Also, in the inward-facing neutral amino acid transporter ASCT2 (*Garaeva et al., 2018*; *Garaeva et al., 2019*), the corresponding HP2/TM8a regions remain mostly rigid during gating, and only the HP2 tip moves to open the binding site or accommodate an inhibitor. Together, these structures suggest that the transport domain pivots around the HP2/TM8a region near resides corresponding to A364 in Glt$_{Ph}$ to open the substrate-binding site. This might be a shared feature of the glutamate transporter family. These movements rely on the remarkable conformational plasticity of HP2 and the interface between the transport and scaffold domains, which differ in each functional intermediate of the transporter. Our recent studies suggest that both translocation of the transport domain and substrate release into the cytoplasm are slow

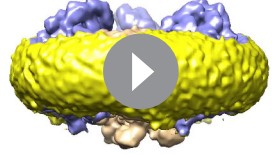

Glt$_{Ph}^{OFS}$-TBOA nanodisc complex

**Video 3.** Transport domain movements coupled to nanodisc distortions.
https://elifesciences.org/articles/58417#video3

processes (*Oh and Boudker, 2018*; *Huysmans et al., 2020*). Most strikingly, subtle packing mutations in HP2 at sites distant from the substrate-binding site decrease affinity in the OFS and IFS and increase the elevator transitions frequency (*Huysmans et al., 2020*).

Our structures show that $Na^+$ ions and L-asp release require movement of the transport domain, mediated by conformational changes of HP2 and the HP2/TM8a-scaffold interface. These extensive conformational changes, involving repacking the domain interface, may explain why substrate gating is slow in the IFS (*Oh and Boudker, 2018*). Gating in the OFS, where only HP2 moves to bind $Na^+$ ions and L-asp is faster (*Hänelt et al., 2015*), although slow HP2 opening has also been proposed (*Riederer and Valiyaveetil, 2019*). Notably, kinetic studies showed that the release (and binding) of one $Na^+$ ion in the IFS, most likely Na2, is rapid (*Oh and Boudker, 2018*). Thus, it is likely that the release of Na2 requires little structural change, limited at most to the change observed in the $Glt_{Ph}^{IFS}$-TBOA structure. Our structural data further suggest that mutations in HP2 may increase the substrate dissociation rate in the IFS by increasing the dynamics of the hairpin and the hairpin/scaffold interface.

Single-molecule studies of the elevator dynamics showed that the rate-limiting high-energy transition state most likely structurally resembles the IFS, and the transport domain might make multiple attempts to achieve a stable observable IFS (*Huysmans et al., 2020*). These studies suggest that multiple IFS conformations exist and are separated by significant energetic barriers. While our structures most likely represent the lower-energy states populated during Cryo-EM imaging, and not the high-energy transition states, their multiplicity supports the existence of a complex inward-facing conformational ensemble.

Significant alterations of the structure of the surrounding membranes and some of the well-structured annular lipids accompany the observed large-scale functional domain movements. In general, it appears that lipids occupy all indentations and crevices on the surface of the protein open to the bilayer and large enough to accommodate hydrocarbon chains even in the absence of specific interactions between the headgroups and protein moieties. The density for some of the lipids, such as $Lipid_{In}$ (*Figure 5a* and *Figure 5—figure supplement 1*), is very well resolved. These lipids display conserved locations and structures in all resolved protein complexes. However, it is unclear whether they are structurally immobilized or exchange rapidly with the surrounding bulk lipids.

Other lipids would have to move in and out of their binding sites during the transport cycle. These include $Lipid_{Out}$, observed in the OFS and IFS, and the structurally symmetric cytoplasmic $Lipid_{Window}$ observed in the IFS. Interestingly, $Lipid_{Out}$ sits between the HP2 N-terminal arm and the scaffold in both the OFS and IFS. Thus, there is an interplay between HP2 and lipids in the two states. In the OFS, when HP2 closes over the binding site, $Lipid_{Out}$ fills the space between the hairpin and the scaffold, and when HP2 opens, it displaces the lipid and interacts directly with the scaffold. In the IFS, $Lipid_{Window}$ moves in when the transport domain leans away from HP2 to open the substrate-binding site and moves out when it closes in. Such intimate involvement of lipids suggests that they can regulate both substrate-binding and elevator dynamics. However, only modest effects of specific lipids on $Glt_{Ph}$ transport activity have been reported thus far (*McIlwain et al., 2015*). Interestingly, mammalian EAAT1 and ASCT2 feature a similar space between the N-terminal arm of HP2 and the scaffold in the OFS and IFS (*Canul-Tec et al., 2017*; *Yu et al., 2019*; *Garaeva et al., 2018*), and likely can accommodate lipids. Further studies are needed to establish the relevance of the identified lipid-binding sites to lipid-mediated regulation reported in mammalian EAATs (*Zerangue et al., 1995*; *Tzingounis et al., 1998*; *Fairman et al., 1998*).

Our structures, $Glt_{Tk}$ structures in nanodiscs, and molecular dynamics simulations, visualize lipid bilayer bending, accommodating the conformational change from the OFS to IFS (*Zhou et al., 2019*; *Arkhipova et al., 2020*). Due to the limited size of the nanodiscs, structural studies do not resolve the long-range effects on the membrane deformations. However, simulations showed that the membrane perturbation extends to nearly 100 Å. The computational study also suggests that the energy penalty of bilayer bending might be as large as 6–7 kcal/mol protomer. Our results show that not only the OFS to IFS transitions but also substrate gating in the IFS involve changes in membrane deformation. Thus, high energetic costs of membrane bending might accompany the glutamate transporter functional cycle, suggesting that the physical properties of lipid bilayers, such as thickness and stiffness (*Lundbaek et al., 2010*; *Bruno et al., 2013*; *Rusinova et al., 2014*), can significantly impact function.

# Materials and methods

## Key resources table

| Reagent type (species) or resource | Designation | Source or reference | Identifiers | Additional information |
|---|---|---|---|---|
| Biological sample (*Escherichia coli*) | DH10B | Invitrogen | | Cells for Glt$_{Ph}$ expression |
| Biological sample (*Escherichia coli*) | BL21(DE3) | Stratagene | | Cells for MSP1E3 expression |
| Recombinant DNA reagent | Glt$_{Ph}$ | DOI: 10.1038/nature03018 | | |
| Recombinant DNA reagent | MSP1E3 | Addgene https://www.addgene.org/20064/ | PRID:Addgene_20064 | |
| Software, algorithm | Origin | OriginLab | | |
| Software, algorithm | Leginon | doi:10.1016/j.jsb.2005.03.010 | | |
| Software, algorithm | Relion | doi:10.7554/eLife.42166 | RRID:SCR_016274 | |
| Software, algorithm | MotionCorr2 | doi:10.1038/nmeth.4193 | | |
| Software, algorithm | CTFFIND4 | doi:10.1016/j.jsb.2015.08.008 | RRID:SCR_016732 | |
| Software, algorithm | UCSF chimera | doi:10.1002/jcc.20084 | RRID:SCR_004097 | |
| Software, algorithm | ResMap | doi:10.1038/nmeth.2727 | | |
| Software, algorithm | Pymol | Schrödinger | RRID:SCR_000305 | |
| Software, algorithm | NanoAnalyze | TAinstruments | | |
| Software, algorithm | Nano ITCRun | TAinstruments | | |
| Software, algorithm | Appion | doi: 10.1016/j.jsb.2009.01.002 | RRID:SCR_016734 | |
| Software, algorithm | PDBePISA | doi:10.1016/j.jmb.2007.05.022 | RRID:SCR_015749 | |
| Software, algorithm | DoGpicker | doi:10.1016/j.jsb.2009.01.004 | | |

## Glt$_{Ph}$ expression, purification, and crosslinking

The fully functional seven-histidine mutant of Glt$_{Ph}$ that has been used in previous studies and that is referred to as wildtype (WT) for brevity, and the K55C/C321A/A364C Glt$_{Ph}$ mutant were expressed as C-terminal His$_8$ fusions and purified as described previously (*Yernool et al., 2004*). Briefly, the plasmids were transformed into *E. coli* DH10-B cells (*Invitrogen*). Cells were grown in LB media supplemented with 0.2 mg/L of ampicillin (*Goldbio*) at 37°C until OD$_{600}$ of 1.0. Protein expression was induced by adding 0.2% arabinose (*Goldbio*) for 3 hr at 37°C. The cells were harvested by centrifugation and re-suspended in 20 mM Hepes, pH 7.4, 200 mM NaCl, 1 mM L-asp, 1 mM EDTA. The suspended cells were broken using Emulsiflex C3 high pressure homogenizer (*Avestin Inc*) in the presence of 0.5 mg/mL lysozyme (*Goldbio*) and 1 mM phenylmethanesulfonyl fluoride (PMSF, *MP Biomedicals*). After centrifugation for 15 min at 5000 g at 4°C to remove the debris, membranes were pelleted by centrifugation at 125,000 g for 60 min. The membranes were homogenized in 20 mM Hepes, pH 7.4, 200 mM NaCl, 1 mM L-asp, 10 mM EDTA, 10% sucrose and pelleted again by centrifugation at 125,000 g for 60 min. The washed membranes were collected and solubilized in Buffer A, containing 20 mM Hepes, pH7.4, 200 mM NaCl, 1 mM L-asp, supplemented with 40 mM n-dodecyl-β-D-maltopyranoside (DDM, *Anatrace, Inc*) at 8 mL per gram of membranes for 2 hr at 4°C. The mixture was clarified by ultracentrifugation for 60 min at 125,000 g, the supernatant was incubated with Ni-NTA resin (*Qiagen*) pre-equilibrated in buffer A with gentle shaking for 2 hr at 4°C. The resin was washed with 5 volumes of Buffer A with 1 mM DDM and 25 mM imidazole, the protein was eluted in the same buffer containing 250 mM imidazole. The eluted protein was concentrated using concentrators with 100 kDa MW cutoff (*Amicon*). The (His)$_8$-tag was cleaved by thrombin (*Sigma*) using 20 U per 1 mg Glt$_{Ph}$ in the presence of 5 mM CaCl$_2$ at room temperature overnight. The reaction was stopped by addition of 10 mM EDTA and 1 mM PMSF. For the WT Glt$_{Ph}$, the protein was further purified by size exclusion chromatography (SEC) in buffer A and 1 mM DDM. The eluted protein was concentrated and used immediately for nanodisc reconstitution. After affinity chromatography and (His)$_8$-tag removal, prior to crosslinking, the K55C/C321A/A364C mutant

protein was reduced with 5 mM Tris(2-carboxyethyl)phosphine (TCEP) at room temperature for 1 hr. Protein was then exchanged into buffer A with 1 mM DDM, using filters (Amico, Inc) with a molecular weight cutoff of 100 kDa. Reduced K55C/C321A/A364C $Glt_{Ph}$ at concentrations below 1 mg/mL was incubated with 10-fold molar excess of $HgCl_2$ for 15 min at room temperature. The protein was concentrated to under 1 ml and purified by SEC in buffer A supplemented with 1 mM DDM. The elution peak fractions were collected and concentrated. The protein concentration was determined by UV absorbance at 280 nm using extinction coefficient of 57,400 $M^{-1}$ $cm^{-1}$ and MW of 44.7 kDa. To check availability of free thiols after crosslinking, proteins were incubated with 5-fold molar excess of fluoroscein-5-maleimide (F5M). Fluorescent F5M-labeled proteins were imaged on SDS-PAGE under blue illumination and stained with Coomassie blue.

## Reconstitution of $Glt_{Ph}$ into nanodiscs

Membrane scaffold protein MSP1E3 (*Denisov et al., 2004*) was expressed and purified from *E. coli* and $Glt_{Ph}$ was reconstituted into lipid nanodiscs as previously described, with modifications (*Ritchie et al., 2009*). Briefly, *E. coli* polar lipid extract and egg phosphatidylcholine in chloroform (*Avanti*) were mixed at 3:1 (w:w) ratio and dried on rotary evaporator and under vacuum overnight. The dried lipid film was resuspended in buffer containing 20 mM Hepes/Tris, pH 7.4, 200 mM NaCl, 1 mM L-asp and 80 mM DDM by 10 freeze/thaw cycles resulting in 20 mM lipid stock. The purified $Glt_{Ph}$ protein in DDM was mixed with MSP1E3 and lipid stock at 0.75:1:50 molar ratio at the final lipid concentration of 5 mM and incubated at 21°C for 30 min. Biobeads SM2 (*Bio-Rad*) were added to one third of the reaction volume and the mixture was incubated at 21°C for 2 hr on a rotator. Biobeads were replaced and incubated at 4°C overnight. The sample containing $Glt_{Ph}^{IFS}$ reconstituted into the nanodiscs in the presence of 1 mM L-asp was cleared by centrifugation at 100,000 g and purified by SEC using a Superose 6 Increase 10/300 GL column (GE Lifesciences) pre-equilibrated with buffer containing 20 mM Hepes/Tris, pH 7.4, 200 mM NaCl and 1 mM L-asp. The peak fractions corresponding to $Glt_{Ph}^{IFS}$-containing nanodiscs were collected for Cryo-EM imaging. To prepare substrate-free WT $Glt_{Ph}$ and $Glt_{Ph}^{IFS}$ in nanodiscs, the reconstitution mixtures were cleared by centrifugation at 100,000 g, diluted with 10 x volume of buffer containing 20 mM Hepes/Tris, pH 7.4, and 50 mM choline chloride, and concentrated using 100 kDa cutoff concentrator. After repeating the procedure twice, substrate-free transporters in nanodiscs were purified by SEC in the same buffer. The peak fractions were collected and immediately supplemented with buffers containing 200 mM NaCl and 10 mM DL-TBOA, 200 mM NaCl and 10 mM TFB-TBOA, or 200 mM NaCl. The presence of the MSP1E3 and $Glt_{Ph}$ proteins in the samples was confirmed by SDS-PAGE. Negative staining electron microscopy was used to confirm the formation and the homogeneity of the nanodisc samples.

## Cryo-EM data collection

To prepare cryo-grids, 3.5 µL of $Glt_{Ph}$-containing nanodiscs (7 mg/mL) supplemented with 1.5 mM fluorinated Fos-Choline-8 (*Anatrace*) were applied to a glow-discharged UltrAuFoil R1.2/1.3 300-mesh gold grid (*Quantifoil*) and incubated for 20 s under 100% humidity at 15°C. Grids were blotted for 2 s and plunge frozen in liquid ethane using Vitrobot Mark IV (*Thermo Fisher Scientific*). For the WT $Glt_{Ph}$ in the presence of DL-TBOA ($Glt_{Ph}^{OFS}$-TBOA), $Glt_{Ph}^{IFS}$ in the presence of TFB-TBOA ($Glt_{Ph}^{IFS}$-TFB-TBOA), and $Glt_{Ph}^{IFS}$ in the presence of 200 mM $Na^+$ ions only ($Glt_{Ph}^{IFS}$-NaCl), the Cryo-EM imaging data were acquired using a Titan Krios microscope (*Thermo Fisher Scientific*) at New York Structural Biology Center operated at 300 kV with a K2 Summit detector with a calibrated pixel size of 1.07325 Å/pixel. A total dose of 68.55 $e^-/Å^2$ ($Glt_{Ph}^{OFS}$-TBOA, $Glt_{Ph}^{IFS}$-NaCl), or 68.70 $e^-/Å^2$ ($Glt_{Ph}^{IFS}$-TFB-TBOA) distributed over 45 frames (1.52 $e^-/Å^2$/frame) was used with an exposure time of 9 s (200 ms/frame) and a defocus range of −1.5 µm to −2.5 µm. For $Glt_{Ph}^{IFS}$ in the presence of DL-TBOA ($Glt_{Ph}^{IFS}$-TBOA), Cryo-EM imaging data were acquired on a Titan Krios microscope at New York Structural Biology Center operated at 300 kV with a K2 Summit detector with a calibrated pixel size of 1.07325 Å/pixel. A total dose of 69.70 $e^-/Å^2$ distributed over 50 frames (1.52 $e^-/Å^2$/frame) was used with an exposure time of 10 s (200 ms/frame) and a defocus range of −1.5 µm to −2.5 µm. For the $Glt_{Ph}^{IFS}$ in the presence of L-asp ($Glt_{Ph}^{IFS}$-Asp), micrographs were acquired on a Titan Krios microscope at New York Structural Biology Center operated at 300 kV with a K2 Summit detector, using a slid width of 20 eV on a GIF Quantum energy filter with a calibrated pixel size of 1.0605 Å/pixel. A total dose of 69.30 $e^-/Å^2$ distributed over 45 frames (1.54 $e^-/Å^2$/frame) was used with an exposure time of 9 s (200 ms/

frame) and defocus range of −1.5 μm to −2.5 μm. For all samples, automated data collection was carried out using Leginon (*Suloway et al., 2005*).

## Image processing

The frame stacks were motion corrected using MotionCorr2 (*Zheng et al., 2017*) and contrast transfer function (CTF) estimation was performed using CTFFIND4 (*Rohou and Grigorieff, 2015*). All further processing steps were done using RELION 3.0 ($Glt_{Ph}^{IFS}$-Asp, $Glt_{Ph}^{IFS}$-TBOA, $Glt_{Ph}^{IFS}$-TFB-TBOA, $Glt_{Ph}^{IFS}$-NaCl) or Relion 3.1 ($Glt_{Ph}^{IFS}$-TBOA) unless otherwise indicated (*Zivanov et al., 2018*). DoGpicker (*Voss et al., 2009*) as part of the Appion processing package (*Lander et al., 2009*) was used for reference-free particle picking. Picked particles were then extracted and subjected to 2D classification to generate 2D class-averages which were used as templates for automated particle picking in Relion. The particles were extracted using a box size of 275 Å with 2x binning and subjected to 2 rounds of 2D classification ignoring CTFs until the first peak.

For $Glt_{Ph}^{IFS}$-Asp, $Glt_{Ph}^{IFS}$-TBOA, $Glt_{Ph}^{IFS}$-TFB-TBOA, and for the $Glt_{Ph}^{OFS}$-TBOA, particles selected from 2D classification were re-extracted without binning and further classified into six classes without enforcing symmetry using initial models generated in CryoSPARC (*Punjani et al., 2017*) and filtered to 40 Å. Particles from the best classes showing trimeric transporter arrangements were subjected to 3D refinement applying C3 symmetry. After conversion, the refinement was continued with a mask excluding the nanodisc. To further improve the resolution of the maps, the particles after 3D refinement were subject to an additional round of 3D classification without alignment with C3 symmetry applying a mask to exclude the nanodisc. Particles from the best class were subjected to further masked refinement and CTF refinement. A masked refinement following CTF refinement yielded final maps with the following resolution: 3.05 Å ($Glt_{Ph}^{IFS}$-Asp), 3.71 Å ($Glt_{Ph}^{IFS}$-TFB-TBOA), 3.39 Å ($Glt_{Ph}^{IFS}$-TBOA), 3.66 Å ($Glt_{Ph}^{OFS}$-TBOA). The resolution limits of the refined maps were assessed using Relion postprocessing and gold standard FSC value 0.143 using masks that excluded the nanodiscs. To search for potential conformational heterogeneity, we also processed these datasets with no symmetry applied at any stage of data processing (C1). The obtained C1 maps showed slightly lower resolution but no detectable difference when compared to the results from the C3 refinement. We also processed all datasets with symmetry expansion (C3) followed by focused 3D classification on one $Glt_{Ph}$ subunit (explained in detail for $Glt_{Ph}^{IFS}$-Na data processing below) and did not find additional conformations. The cryo-EM map of $Glt_{Ph}^{IFS}$-TBOA was processed using the RESOLVE density modification program implemented in Phenix, which improved the overall estimated resolution by 0.01 Å and enabled slightly better visualization of the density of the bound TBOA benzyl group (*Terwilliger et al., 2020*; *Afonine et al., 2010*).

During processing of the data for $Glt_{Ph}^{IFS}$-NaCl, 529,155 particles selected from 2D classification were re-extracted without binning and were subjected to 3D classification with K = 1 and no symmetry applied, using $Glt_{Ph}^{IFS}$-Asp map as the initial model. The same particles were subject to 3D refinement with C3 symmetry. After conversion, the refinement was continued with a mask to exclude the nanodisc, resulting in a 3.56 Å resolution map. To probe for conformational heterogeneity, we performed symmetry expansion implemented in Relion (*Scheres, 2016*). 1,587,465 protein subunits were rotated to the same position and subjected to a focused 3D classification without alignment with T = 40 into 10 classes. The local mask was generated using Chain A of PDB model 3KBC (*Reyes et al., 2009*) and included only densities from one subunit of the reference map. Two different conformations were observed. From the 10 classes, five classes showed a conformation identified as $Glt_{Ph}^{IFS}$-Na and five classes showed a different conformation identified as $Glt_{Ph}^{IFS}$-Apo-open. The best $Glt_{Ph}^{IFS}$-Na class (191,349 particles), which contained 12% of the symmetry expanded protomers and the best $Glt_{Ph}^{IFS}$-Apo-open class (148,582 particles), which contained 9% of the symmetry expanded particles, were separately subjected to a final focused 3D refinement with C1 using a mask to exclude the nanodisc. The local angular searches in this refinement were conducted only around the expanded set of orientations to prevent contributions from the neighbor subunits in the same particle. The resulting maps were postprocessed in Relion using the same mask as in 3D classification after symmetry expansion. The final resolution at gold standard FSC value 0.143 was estimated as 3.52 Å for the $Glt_{Ph}^{IFS}$-Apo-open map and 3.66 Å for $Glt_{Ph}^{IFS}$-Na map. Local resolution variations were estimated using ResMap (*Kucukelbir et al., 2014*). After symmetry expansion with C3, we also tried to first subtract the density outside of one $Glt_{Ph}$ subunit and then perform 3D

classification without alignment on the subtracted particles. The signal subtraction did not further improve the 3D classification and the 3D refinement.

## Model building and refinement

For atomic model building from $Glt_{Ph}^{IFS}$-Asp, $Glt_{Ph}^{IFS}$-TBOA, and $Glt_{Ph}^{IFS}$-TFB-TBOA maps, crystal structure of $Glt_{Ph}$ in the IFS (PDB code 3KBC) (**Reyes et al., 2009**) was docked into the density maps using UCSF Chimera (**Pettersen et al., 2004**). For the WT $Glt_{Ph}^{OFS}$-TBOA, crystal structure of $Glt_{Ph}$ in the OFS (PDB code 2NWW) (**Boudker et al., 2007**) was docked into the density. For $Glt_{Ph}^{IFS}$-Na or $Glt_{Ph}^{IFS}$-Apo-open, one subunit of 3KBC was docked into the density. After the first rounds of the real-space refinement using Phenix (**Afonine et al., 2010**), miss-aligned regions were manually rebuilt and missing side chains and residues were added in COOT (**Emsley et al., 2010**). 1-palmitoyl-2-oleoyl-sn-glycero-3-phosphoethanolamine (POPE) was used as a model lipid and placed into the excess densities which resembled lipid molecules. The acyl chains or ethanolamine heads were truncated to fit the visible densities. Models were iteratively refined applying secondary structure restraints and validated using Molprobity (**Chen et al., 2010**). For further cross validation and to check for overfitting, all atoms of each model were randomly displaced by 0.3 Å and each resulting model was refined against the first half-map obtained from processing. FSC between the refined models and the half-maps used during the refinement were calculated and compared to the FSC between the refined models and the other half-maps. In addition, the FSC between the refined model and sum of both half-maps was calculated. The resulting FSC curves were similar showing no evidence of overfitting.

## Isothermal titration calorimetry

For ITC experiments, $Glt_{Ph}$ K55C/C321A/A364C proteins were purified by affinity chromatography as above. After (His)$_8$-tag removal, prior to crosslinking, the K55C/C321A/A364C protein was reduced with 5 mM Tris(2-carboxyethyl)phosphine (TCEP) at room temperature for 1 hr. Protein was then exchanged into buffer A with 1 mM DDM using filters (Amico, Inc) with a molecular weight cut-off of 100 kDa. Reduced K55C/C321A/A364C $Glt_{Ph}$ at concentrations below 1 mg/mL was incubated with 10-fold molar excess of $HgCl_2$ for 15 min at room temperature. The protein was concentrated, diluted with 10 x volume of substrate-free buffer containing 20 mM Hepes/Tris, pH 7.4, 50 mM choline chloride and 1 mM DDM, and re-concentrated. After repeating the procedure twice, the protein was purified by SEC in the same buffer. Protein samples at 40 µM in substrate-free buffer supplemented with 200 mM NaCl, were loaded into the reaction cell of an Affinity ITC (TA Instruments, Inc). The injection syringe was loaded with a solution containing 20 mM Hepes/Tris, pH 7.4, 50 mM choline chloride, 200 mM NaCl, 400 µM TFB-TBOA or DL-TBOA. Titrant aliquots of 2 µL were injected every 5 min at 15°C. Binding isotherms were fitted to independent binding site model using NanoAnalyze software (TA Instruments, Inc).

## Acknowledgements

The authors thank members of the Boudker lab for helpful discussions. Drs Biao Qiu, Maria E Falzone, and Jan Rheinberger for helpful suggestions on cryo-EM data processing, Dr. Krishna Reddy for help on ITC experiment, and R Lea Sanford for insightful discussions. This work was supported by NIH Grants R01NS064357 and R37NS085318 (to OB). All EM data collections were carried out at the Simons Electron Microscopy Center and National Resource for Automated Molecular Microscopy located at the New York Structural Biology Center, supported by grants from the Simons Foundation (349247), NYSTAR, and the NIH National Institute of General Medical Sciences (GM103310) with additional support from Agouron Institute (F00316) and NIH S10 OD019994-01. Initial negative stain screening was performed at the Weill Cornell Microscopy and Image Analysis Core Facility, with the help of Dr. L Cohen-Gould.

## Additional information

### Competing interests

Olga Boudker: Senior editor, *eLife*. The other author declares that no competing interests exist.

## Funding

| Funder | Grant reference number | Author |
|---|---|---|
| National Institutes of Health | R37NS085318 | Olga Boudker |
| National Institutes of Health | R01NS064357 | Olga Boudker |

The funders had no role in study design, data collection and interpretation, or the decision to submit the work for publication.

## Author contributions

Xiaoyu Wang, Conceptualization, Data curation, Formal analysis, Validation, Investigation, Methodology, Writing - original draft, Writing - review and editing; Olga Boudker, Conceptualization, Resources, Data curation, Formal analysis, Supervision, Funding acquisition, Methodology, Writing - original draft, Project administration, Writing - review and editing

## Author ORCIDs

Xiaoyu Wang (iD) https://orcid.org/0000-0002-8745-8238
Olga Boudker (iD) https://orcid.org/0000-0001-6965-0851

## Decision letter and Author response

Decision letter https://doi.org/10.7554/eLife.58417.sa1
Author response https://doi.org/10.7554/eLife.58417.sa2

# Additional files

## Supplementary files

• Transparent reporting form

## Data availability

Cryo-EM coordinate files and electron density maps have been deposited in PDB under the following codes: GltPh OFS-TBOA: PDB 6X17, EMD-21991 GltPh IFS-Asp: PDB 6X15, EMD-21989 GltPh IFS-TBOA: PDB 6X16, EMD-21990 GltPh IFS-TFB-TBOA: PDB 6X14, EMD-21988 GltPh IFS-Na: PDB 6X13, EMD-21987 GltPh IFS-Apo-open: PDB 6X12, EMD-21986.

The following datasets were generated:

| Author(s) | Year | Dataset title | Dataset URL | Database and Identifier |
|---|---|---|---|---|
| Wang X, Boudker O | 2020 | Inward-facing sodium-bound state of the glutamate transporter homologue GltPh | http://emsearch.rutgers.edu/atlas/21987_summary.html | EMDataBank, EMD-21987 |
| Wang X, Boudker O | 2020 | Inward-facing state of the glutamate transporter homologue GltPh in complex with TBOA | http://emsearch.rutgers.edu/atlas/21990_summary.html | EMDataBank, EMD-21990 |
| Wang X, Boudker O | 2020 | Outward-facing state of the glutamate transporter homologue GltPh in complex with TBOA | https://www.rcsb.org/structure/6X17 | RCSB Protein Data Bank, 6X17 |
| Wang X, Boudker O | 2020 | Inward-facing state of the glutamate transporter homologue GltPh in complex with TFB-TBOA | http://emsearch.rutgers.edu/atlas/21988_summary.html | EMDataBank, EMD-21988 |
| Wang X, Boudker O | 2020 | Inward-facing Apo-open state of the glutamate transporter homologue GltPh | http://emsearch.rutgers.edu/atlas/21986_summary.html | EMDataBank, EMD-21986 |
| Wang X, Boudker O | 2020 | Inward-facing state of the glutamate transporter homologue GltPh in complex with L-aspartate and sodium ions | https://www.rcsb.org/structure/6X15 | RCSB Protein Data Bank, 6X15 |
| Wang X, Boudker O | 2020 | Inward-facing state of the glutamate transporter homologue | https://www.rcsb.org/structure/6X16 | RCSB Protein Data Bank, 6X16 |

| | | | GltPh in complex with TBOA | | |
|---|---|---|---|---|---|
| Wang X, Boudker O | 2020 | Inward-facing state of the glutamate transporter homologue GltPh in complex with TFB-TBOA | https://www.rcsb.org/structure/6X14 | RCSB Protein Data Bank, 6X14 |
| Wang X, Boudker O | 2020 | Inward-facing sodium-bound state of the glutamate transporter homologue GltPh | https://www.rcsb.org/structure/6X13 | RCSB Protein Data Bank, 6X13 |
| Wang X, Boudker O | 2020 | Inward-facing Apo-open state of the glutamate transporter homologue GltPh | https://www.rcsb.org/structure/6X12 | RCSB Protein Data Bank, 6X12 |
| Wang X, Boudker O | 2020 | Outward-facing state of the glutamate transporter homologue GltPh in complex with TBOA | http://emsearch.rutgers.edu/atlas/21991_summary.html | EMDataBank, EMD-21991 |
| Wang X, Boudker O | 2020 | Inward-facing state of the glutamate transporter homologue GltPh in complex with L-aspartate and sodium ions | http://emsearch.rutgers.edu/atlas/21989_summary.html | EMDataBank, EMD-21989 |

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
