## [Decision Letter]

**Acceptance summary:**

The manuscript by Wang and Boudker details structural investigations on Glt_Ph_, a homologue of human glutamate transporters and a model for transporters operating via an elevator-like alternating-access mechanism. The authors extensively explore the conformational flexibility of Glt_Ph_ in the inward-facing state and in lipid nanodiscs, using a locked variant. They demonstrate comparably large motions of the transport domain depending on binding of substrates, but also more subtle rearrangements in the binding site region. Together, this results in important novel insights in gating and transport. In addition, the paper confirms and extends previous observations for Glt_Tk_ and ASCT2 concerning the role of HP2 in substrate release on the cytoplasmic side of the protein, as well as on extensive bilayer deformations during transport predicted by molecular dynamics simulations indicating a dynamic interplay between the conformational state of elevator-like proteins and their environment.

**Decision letter after peer review:**

Thank you for submitting your article "Large domain movements through lipid bilayer mediate substrate release and inhibition of glutamate transporters" for consideration by *eLife*. Your article has been reviewed by three peer reviewers, and the evaluation has been overseen by a Reviewing Editor an Richard Aldrich as the Senior Editor. The following individual involved in review of your submission has agreed to reveal their identity: Eric R Geertsma (Reviewer #3).

The reviewers have discussed the reviews with one another and the Reviewing Editor has drafted this decision to help you prepare a revised submission.

Summary:

The manuscript by Wang and Boudker details structural investigations on Glt_Ph_, a homologue of human glutamate transporters and a model for transporters operating via an elevator-like alternating-access mechanism. The authors extensively explore the conformational flexibility of Glt_Ph_ in the inward-facing state and in lipid nanodiscs, using a locked variant. They demonstrate comparably large motions of the transport domain depending on binding of substrates, but also more subtle rearrangements in the binding site region. Together, this results in important novel insights in gating and transport. In addition, the paper confirms and extends previous observations for Glt_Tk_ and ASCT2 concerning the role of HP2 in substrate release on the cytoplasmic side of the protein, as well as on extensive bilayer deformations during transport predicted by molecular dynamics simulations indicating a dynamic interplay between the conformational state of elevator-like proteins and their environment.

Essential revisions:

1) A weakness of the study is that the most important findings (conformational changes associated with inhibitor binding and substrate release on the cytoplasmic side) are deduced from structures of a cross-linked triple-cysteine mutant K55C/C321A/A364C, which artificially traps the transporter in an inward-facing state. This choice is justified by the fact that crystal structures of the locked and unconstrained versions had shown a very similar structure (subsection “Large range of motions of the transport domain in the IFS”). However, this type of information is not available for several of the new structures for which no accompanying crystal structure was determined. Therefore, the consequences of this approach need to be discussed in more detail. E.g., it seems that all movements of the transport domain pivot around the 55/364 crosslink (Figure 1). Is this the only mode of flexibility still available due to the crosslink? Are there additional reasons to expect that these are representative conformations?

2) The authors should also comment on the symmetry of these structures, which apparently reflects the cross-linking, even though it is known that the native protein can form asymmetric arrangements, based on e.g. HS-AFM (Ruan et al., 2017), smFRET (Akyuz et al., 2013). Thus, it would be helpful if the authors could explain whether any asymmetry is observed between the subunits in any of the other cryo-EM images – whether in the global conformation or in the orientation of HP2 – and explicitly describe the choice of assuming either C1 or C3 symmetry for each map.

3) The putative density of the TBOA molecule in the Glt_Ph_^IFS^ map (Figure 2—figure supplement Figure 1B) is not sufficiently strong to unambiguously determine the binding mode of the inhibitor. While the authors attribute the absence of density for the benzyl group to a mixture of enantiomers binding to the pocket, this has not been reported to be a problem for other cryo-EM structures (or X-ray structures) of SLC1 members with TBOA. In a previous study of the last author, TBA was used instead of TBOA to circumvent the problems with mixed enantiomer binding; is there a reason why the authors did not choose this approach in the current study? An alternative explanation for the absence of benzyl density for TBOA is that, if not all three protomers are occupied by an inhibitor (see point #1), the density of unoccupied and occupied sites will be averaged when C3 symmetry is applied. A third possibility for the origin of the weak benzyl moiety density, is that the cryo-EM grid contains a mixture of asp- and inhibitor-bound species which are averaged during EM data processing. Is the binding affinity of the two inhibitors TBOA and TFB-TBOA cysteine-crosslinked variant known? If not, and if the authors are able to carry out follow-up experiments, such measurements would help demonstrate that the sites are likely to be saturated. The authors did not describe any measures in the purification protocol to remove aspartate carried over from the media (L-aspartate binds with high affinity) to prevent this scenario. Thus, an alternate suggested experiment would be to increase the purity of the sample in the presence of TBOA (e.g., by extensive washes of the membranes or affinity resin) and repeat the cryo-EM measurements. In the absence of either of these additional experiments, the authors should revise their conclusions from the TBOA-bound inward-facing EM structure.

4) The observed bilayer deformations are very interesting and even mentioned in the Abstract, but somewhat underlit and discussed in only one short paragraph. The authors are encouraged to analyze this data in more detail, e.g., using a plot similar to Figure 2 in Zhou et al., 2019 or Figure 4B in Arkhipova et al., 2020. This would allow to determine, for example, whether the deformations are indeed limited to the transport domains as predicted/observed in the indicated publications.

5) The protein is reconstituted in MSP1E3 nanodiscs. The authors should comment on to what extent the size of the nanodisc is expected to affect the conformation flexibility of the protein. E.g., do they see direct contacts between the transport domain and the nanodisc scaffold protein? Is the outward movement of the transport domain with TFB-TBOA constrained by the nanodisc, either directly (direct contacts) or indirectly (by the nanodisc putting a limitation on the extent of membrane deformation)?

6) The study explicitly discusses the positioning and potential functional relevance of lipids in different regions of the structure. The authors suggest that the annular lipids in Glt_Ph_ play a similar regulatory role to arachidonic acid for the mammalian transporters, yet they do not include any lipid analysis to determine the identity of the annular lipids resolved in the density maps to understand which lipid in bacterial membranes could have a similar regulatory effect on Glt_Ph_. Without knowledge of the lipid identity and functional data assessing whether these lipids modulate the kinetics of transport in reconstituted liposomes, the proposed functional role of lipids in regulating substrate affinity and conformational dynamics remains speculative. Please revise accordingly.

---

## [Author Response]

Essential revisions:1) A weakness of the study is that the most important findings (conformational changes associated with inhibitor binding and substrate release on the cytoplasmic side) are deduced from structures of a cross-linked triple-cysteine mutant K55C/C321A/A364C, which artificially traps the transporter in an inward-facing state. This choice is justified by the fact that crystal structures of the locked and unconstrained versions had shown a very similar structure (subsection “Large range of motions of the transport domain in the IFS”). However, this type of information is not available for several of the new structures for which no accompanying crystal structure was determined. Therefore, the consequences of this approach need to be discussed in more detail. E.g., it seems that all movements of the transport domain pivot around the 55/364 crosslink (Figure 1). Is this the only mode of flexibility still available due to the crosslink? Are there additional reasons to expect that these are representative conformations?

We agree with the reviewers' concern regarding the crosslink's potential structural constraints. However, because the wild-type Glt_Ph_ strongly prefers the outward-facing state (OFS) over the inward-facing state (IFS) in lipid bilayer environment and the presence of Na^+^ ions or TBOA (Akyuz et al., 2015, Ruan et al., 2017, Huang et al., 2020, Georgieva et al., 2013, Hanelt et al., 2013), crosslinking was necessary to image the inhibitor-bound and substrate-releasing IFS. As mentioned by the reviewer and in the paper, the L-Asp-bound crosslinked IFS is nearly identical to the unconstrained IFS (Akyuz et al., 2015, Verdon and Boudker, 2012). We think that the substrate-releasing and inhibitor-bound IF states reported in our study are also minimally affected by the crosslink based on the following evidence:

1) The structure of the Na^+^-bound Glt_Tk_ (which shares a 77% sequence identity with Glt_Ph_) in the IFS (PDB 6XWR) is very similar to our Glt_Ph_^IFS^-Na structure (Arkhipova et al., 2020). The overall RMSD is 0.7 Å, and the crosslink sites superimpose very well (new Figure 3—figure supplement 2C and Author response image 1). The CA-CA distances between 55C and 364C in Glt_Ph_ and the corresponding residues in Glt_Tk_ are 7.6 and 7.2 Å. Furthermore, when we superimposed the trimerization domains of Glt_Tk_ in Na^+^ with our Glt_Ph_^IFS^-Asp (Author response image 1), we again observed nearly identical configurations of the crosslink site, with the CA-CA distance between the cysteines in Glt_Ph_^IFS^-Asp of 7.4 Å. Thus, the crosslinks do not affect the transport domain's position relative to the scaffold in these structures.

**Author response image 1. sa2fig1:** Glt_Ph_^IFS^-Na (a) and Glt_Ph_^IFS^-Asp (b) (in colors) aligned to Glt_Tk_ Na^+^-bound structure (grey, PDB accession code 6XWR) on the trimerization domain. 55C and 364C side chains of Glt_Ph_^IFS^-Na are shown as sticks and CA atoms of the corresponding K57 and A364 of Glt_Tk_ are shown as spheres.

2) In the Glt_Ph_^IFS^-TFB-TBOA structure, the transport domain adopts the same conformation as the transport domain of Glt_Ph_^OFS^-TBOA (Figure 2—figure supplement 1D). This observation suggests that the crosslink does not alter the conformation of the transport domain in Glt_Ph_^IFS^-TFB-TBOA. We also note that in published unconstrained IFS structures of a neutral amino acid transporter ASCT2 (Garaeva et al., 2019), the transport domain swings away from the scaffold in a similar manner to Glt_Ph_^IFS^-TFB-TBOA. Only the tip of HP2 flips open to release the substrate or to bind TBOA. The rest of HP2, including the region corresponding to 55C-364C in Glt_Ph_, remains mostly rigid. The CA-CA distance between A440 and L100 (equivalent to 55C and 364C in Glt_Ph_) in TBOA- and substrate-bound structures are 7.9 and 8.3 Å, reflecting little structural rearrangements. Notably, Glt_Ph_^IFS^-TFB-TBOA and TBOA-bound ASCT2 IFS show similar HP2 opening and “unlocked” transport domain position. In summary, the transport domain of the glutamate transporters can move away from the scaffold domain in the IFS to various degrees, pivoting around the HP2/TM8a contact region, where the crosslink is placed.

We added the above discussion to the Discussion section.

2) The authors should also comment on the symmetry of these structures, which apparently reflects the cross-linking, even though it is known that the native protein can form asymmetric arrangements, based on e.g. HS-AFM (Ruan et al., 2017), smFRET (Akyuz et al., 2013). Thus, it would be helpful if the authors could explain whether any asymmetry is observed between the subunits in any of the other cryo-EM images – whether in the global conformation or in the orientation of HP2 – and explicitly describe the choice of assuming either C1 or C3 symmetry for each map.

As the reviewer pointed out, we could not observe OFS conformations in the crosslinked IFS constructs, based on the ~100 % crosslinking efficiency, confirmed by SDS-PAGE (data not shown). However, we went through an extensive search to classify any conformational heterogeneity possible within the IFS. The initial 3D refinements with C1 symmetry of the particles imaged in 200 mM NaCl showed the density around HP2 that appeared to be a mixture of two conformations. This observation prompted us to perform symmetry expansion, followed by focused 3D classification on one protomer (Figure 1—figure supplement 1B). Using this approach, we identified two states (Glt_ph_^IFS^–Na and Glt_Ph_^IFS^-apo-open). Under these conditions, the trimers are likely asymmetric, with some protomers bound to Na^+^ ions and other not. For all other datasets (Glt_Ph_^OFS^-TBOA, Glt_Ph_^IFS^-Asp, Glt_Ph_^IFS^–TBOA, Glt_Ph_^IFS^–TFB-TBOA), we did not observe structural heterogeneity using this strategy and applied C3 symmetry to achieve the highest resolution of the final maps. The detailed description of data processing strategies and comments on the symmetry are in the subsection “Image processing” in the Materials and methods section.

3) The putative density of the TBOA molecule in the Glt_Ph_^IFS^ map (Figure 2—figure supplement 1B) is not sufficiently strong to unambiguously determine the binding mode of the inhibitor. While the authors attribute the absence of density for the benzyl group to a mixture of enantiomers binding to the pocket, this has not been reported to be a problem for other cryo-EM structures (or X-ray structures) of SLC1 members with TBOA.

To address the concern, we reprocessed our data in Relion 3.1 and were able to improve the resolution from 3.66 Å to 3.39 Å (description is in the revised Materials and methods). The improved density corresponding to TBOA benzyl group enables us to model the bound inhibitor with good confidence (revise Figure 2A, B, C, and the revised text in the subsection “Two transporter blockers bind differently to Glt_Ph_^IFS^”). We note that the ambiguous TBOA densities are a shared problem for the reported cryo-EM and X-ray structures. In the first report on TBOA-bound crystal structure (Boudker et al., 2007), we used brominated TBOA and anomalous scattering to place the benzyl group. In our updated 3.39 Å Glt_Ph_^IFS^ -TBOA map, the excess density for TBOA is as good or better than in previous reports (Garaeva et al., 2019, Arkhipova et al., 2020).

In a previous study of the last author, TBA was used instead of TBOA to circumvent the problems with mixed enantiomer binding; is there a reason why the authors did not choose this approach in the current study?

Unfortunately, TBA is no longer commercially available.

An alternative explanation for the absence of benzyl density for TBOA is that, if not all three protomers are occupied by an inhibitor (see point #1), the density of unoccupied and occupied sites will be averaged when C3 symmetry is applied.

As explained above, we were able to improve the resolution by reprocessing the data.

A third possibility for the origin of the weak benzyl moiety density, is that the cryo-EM grid contains a mixture of asp- and inhibitor-bound species which are averaged during EM data processing. Is the binding affinity of the two inhibitors TBOA and TFB-TBOA cysteine-crosslinked variant known? If not, and if the authors are able to carry out follow-up experiments, such measurements would help demonstrate that the sites are likely to be saturated. The authors did not describe any measures in the purification protocol to remove aspartate carried over from the media (L-aspartate binds with high affinity) to prevent this scenario. Thus, an alternate suggested experiment would be to increase the purity of the sample in the presence of TBOA (e.g., by extensive washes of the membranes or affinity resin) and repeat the cryo-EM measurements. In the absence of either of these additional experiments, the authors should revise their conclusions from the TBOA-bound inward-facing EM structure.

To address the concern, we measured the binding affinity of TBOA and TFB-TBOA to the Hg-crosslinked Glt_Ph_ 55C/321A/364C IFS construct by ITC, in the presence of 200 mM NaCl (used in Cryo-EM imaging). The affinities are 6.6 and 3.8 μM, respectively (new Figure 2—figure supplement 2, and the revised text in the subsection “Two transporter blockers bind differently to Glt_Ph_^IFS^”). Because we used 10 mM DL-TBOA or TFB-TBOA in our imaging buffers, the sites were likely saturated.

We also think that it is very unlikely that there were any residual L-Asp in our Cryo-EM samples. The L-Asp-free transporters were prepared by SEC of nanodisc-reconstituted crosslinked transporters in Na^+^-free buffer (Materials and methods subsection “Reconstitution of Glt_Ph_ into nanodiscs”). Under these conditions, the L-Asp affinity for the transporter is very low. We and others have employed similar protocols to prepare substrate-free transporter samples in the past with success (Reyes et al., 2013). NaCl and inhibitors were then added to the substrate-free transporter before freezing grids.

4) The observed bilayer deformations are very interesting and even mentioned in the Abstract, but somewhat underlit and discussed in only one short paragraph. The authors are encouraged to analyze this data in more detail, e.g., using a plot similar to Figure 2 in Zhou et al., 2019 or Figure 4B in Arkhipova et al., 2020. This would allow to determine, for example, whether the deformations are indeed limited to the transport domains as predicted/observed in the indicated publications.

We thank the reviewer for the excellent suggestion to analyze the deformation of the nanodisc in more detail. Accordingly, we remade Figure 5, adding additional panels and analyzed the membrane deformation as suggested by the reviewer. Corresponding edits to the text are in the subsection “Transport domain movements coupled to lipid bilayer”.

5) The protein is reconstituted in MSP1E3 nanodiscs. The authors should comment on to what extent the size of the nanodisc is expected to affect the conformation flexibility of the protein. E.g., do they see direct contacts between the transport domain and the nanodisc scaffold protein? Is the outward movement of the transport domain with TFB-TBOA constrained by the nanodisc, either directly (direct contacts) or indirectly (by the nanodisc putting a limitation on the extent of membrane deformation)?

The reviewers raise an interesting point. In Glt_Ph_^IFS^-TFB-TBOA, the edge of the transport domain comes very close to the nanodisc rim, as is evident in the cytoplasmic view (Figure 5E). Notably, we first reconstituted crosslinked Glt_Ph_^IFS^ into nanodiscs in the presence of Na^+^ and L-Asp. After substrate removal (see above and Materials and methods), we added Na^+^ and TFB-TBOA back to the nanodisc sample. Therefore, our study shows that after reconstitution into nanodisc, the transport domain can swing out to accommodate TFB-TBOA. We do not know if without the constrain of the nanodisc, the transport domain could swing out even more. We have added a corresponding sentence to the subsection “Transport domain movements coupled to lipid bilayer”.

6) The study explicitly discusses the positioning and potential functional relevance of lipids in different regions of the structure. The authors suggest that the annular lipids in Glt_Ph_ play a similar regulatory role to arachidonic acid for the mammalian transporters, yet they do not include any lipid analysis to determine the identity of the annular lipids resolved in the density maps to understand which lipid in bacterial membranes could have a similar regulatory effect on Glt_Ph_. Without knowledge of the lipid identity and functional data assessing whether these lipids modulate the kinetics of transport in reconstituted liposomes, the proposed functional role of lipids in regulating substrate affinity and conformational dynamics remains speculative. Please revise accordingly.

We have revised the text, removing speculations on the possible regulatory role of the lipids in the Discussion.